# A *mex3* homolog is required for differentiation during planarian stem cell lineage development

Shu Jun Zhu[1,2], Stephanie E Hallows[1], Ko W Currie[1,2], ChangJiang Xu[3], Bret J Pearson[1,2,4]*

[1]Program in Developmental and Stem Cell Biology, The Hospital for Sick Children, Toronto, Canada; [2]Department of Molecular Genetics, University of Toronto, Toronto, Canada; [3]Terrence Donnelly Centre for Cellular and Biomedical Research, Toronto, Canada; [4]Ontario Institute for Cancer Research, Toronto, Canada

**Abstract** Neoblasts are adult stem cells (ASCs) in planarians that sustain cell replacement during homeostasis and regeneration of any missing tissue. While numerous studies have examined genes underlying neoblast pluripotency, molecular pathways driving postmitotic fates remain poorly defined. In this study, we used transcriptional profiling of irradiation-sensitive and irradiation-insensitive cell populations and RNA interference (RNAi) functional screening to uncover markers and regulators of postmitotic progeny. We identified 32 new markers distinguishing two main epithelial progenitor populations and a planarian homolog to the MEX3 RNA-binding protein (*Smed-mex3-1*) as a key regulator of lineage progression. *mex3-1* was required for generating differentiated cells of multiple lineages, while restricting the size of the stem cell compartment. We also demonstrated the utility of using *mex3-1(RNAi)* animals to identify additional progenitor markers. These results identified *mex3-1* as a cell fate regulator, broadly required for differentiation, and suggest that *mex3-1* helps to mediate the balance between ASC self-renewal and commitment.

*For correspondence: bret.pearson@utoronto.ca

Competing interests: The authors declare that no competing interests exist.

## Introduction

Adult stem cells (ASCs) are ultimately responsible for all tissue turnover in humans, which has been estimated to be approximately $10^{10}$ cells per day (*Reed, 1999*). This feat is achieved through a delicate balance of proliferation and differentiation, in order to maintain a stable stem cell population while replacing the exact number and type of cells lost to cell turnover or injury. This requires inherent asymmetry in stem cell lineages, with some daughter cells retaining stem cell identity while others become committed to differentiate (*Rambhatla et al., 2001*; *Sherley, 2002*; *Simons and Clevers, 2011*). Asymmetry in cell fate outcomes in stem cell lineages is known to happen in several ways. Asymmetry can be largely intrinsic, driven by the asymmetric distribution of RNA and proteins that drive different fates (*Bossing et al., 1996*; *Doe, 1996*, *2008*; *Doe and Bowerman, 2001*; *Bayraktar et al., 2010*). For example, in *Drosophila* neuroblasts, the cell fate determinant Prospero is physically segregated into the daughter cell of a neuroblast division, where it drives differentiation and suppresses stem cell identity (*Doe et al., 1991*). In contrast, the size of the stem cell population can be controlled almost entirely by extrinsic means, such as in the mammalian intestinal crypt where paneth cells use WNT/Lgr5 signaling to maintain stem cell identity (*Snippert et al., 2010*; *Sato et al., 2011*). As the paneth cell niche expands in colon cancer, so too does the stem cell population (*de Lau et al., 2007*). Other stem cell types can use a combination of mechanisms, such as in the mammalian postnatal cortex of the brain where Hedgehog signaling maintains stem cell identity, and asymmetric segregation of RNA-binding protein complexes and cellular processes determines cell fate choice

**eLife digest** Adult tissues constantly replace the millions of cells they lose on a daily basis. This is made possible by adult stem cells. But how is a stable population of stem cells maintained throughout the life of the organism with constant cell division? One way this can be accomplished is if at every stem cell division, only one of the daughter cells remains a stem cell, while the other becomes specialized. For humans, if this balance is disturbed, cancers may result from too many stem cells, and early aging may result from too few stem cells.

A freshwater flatworm called *Schmidtea mediterranea* is known for its ability to regenerate nearly every part of its body after injury. This flatworm possesses stem cells called neoblasts that can form all of the flatworm's different cell types both during regeneration and during normal tissue turnover. Evidence suggests that the number of neoblasts and the number of specialized cells that neoblasts produce are finely balanced, similar to adult human tissues. However, little is known about the mechanism that controls whether a neoblast takes on a more specialized form.

To express a gene, it must first be copied or 'transcribed' into an RNA molecule. Identifying the RNA molecules that are enriched in the non-stem cells that develop from neoblasts could therefore indicate which genes regulate the cell specialization process. These RNA molecules could also be used as markers that identify which cells have taken on a more specialized form. Using techniques called transcriptional profiling and RNA interference, Zhu et al. identified 32 new markers that indicate that the neoblasts have started to specialize into epithelial cells: cells that line the surfaces of many structures in the body. Further investigation revealed that one gene, called *mex3-1*, is needed for many specialized cell types—not just epithelial cells—to mature from neoblasts in the flatworms. In doing so, *mex3-1* also limits the size of the stem cell population.

Equivalents of *mex3-1* are found in many different species including humans, and so Zhu et al.'s results may help us to understand how other animals regenerate and control the size of their stem cell populations. Mutant flatworms that cannot express *mex3-1* could also be used to study other genes that help neoblasts to specialize.

(*Machold et al., 2003*; *Miller and Gauthier-Fisher, 2009*; *Vessey et al., 2012*). For both regenerative medicine and cancer biology, elucidating how non-stem cell fates are specified is a fundamental aspect of understanding the mechanisms of stem cell lineage development.

The freshwater planarian *Schmidtea mediterranea* (a Lophotrochozoan flatworm) is quickly becoming a powerful model system to study gene function, regeneration, and ASC biology (*Salo and Baguna, 2002*; *Sánchez Alvarado, 2003, 2006*; *Cebria, 2007*; *Gurley and Sánchez Alvarado, 2008*; *Rossi et al., 2008*; *Salo et al., 2009*; *Aboobaker, 2011*; *Gentile et al., 2011*; *Tanaka and Reddien, 2011*; *Baguna, 2012*; *Elliott and Sánchez Alvarado, 2013*; *Rink, 2013*). Asexual planarians are constitutive adult animals and their well-known regenerative abilities are dependent on neoblasts, which possess stem cell activity and express the *piwi* homolog *smedwi-1* (*piwi-1*) (*Sánchez Alvarado and Kang, 2005*; *Reddien et al., 2005b*; *van Wolfswinkel et al., 2014*). Neoblasts are broadly distributed throughout the mesenchyme where they constitute approximately 20–30% of all the cells in the animal, and serve to replenish all the differentiated cell types during normal tissue turnover and regeneration after injury (*Krichinskaya and Martynova, 1975*; *Hayashi et al., 2006*; *Wagner et al., 2011*). At steady state, the population of neoblasts is relatively constant in number despite high levels of tissue turnover, indicating that proliferation, self-renewal, and differentiation are finely balanced (*Pellettieri and Sánchez Alvarado, 2007*; *Pearson and Sánchez Alvarado, 2008*; *Pellettieri et al., 2010*). Currently, how neoblasts make the choice to adopt non-stem cell fates remains largely unknown; physical asymmetric segregation of cell fate components has not been demonstrated, and descriptions of any permissive niche-like signal remain elusive (*Reddien, 2013*; *Rink, 2013*).

Although all planarian stem cells are *piwi-1*+, not all *piwi-1*+ cells are necessarily stem cells. Evidence increasingly shows commitment to particular lineages can occur at the *piwi-1*+ level as small numbers of these cells also express tissue-specific genes, though it remains unknown whether these cells are self-renewing or will directly differentiate (*Lapan and Reddien, 2012*; *Cowles et al., 2013*; *Currie and Pearson, 2013*; *Scimone et al., 2014*). Furthermore, recently neoblasts were divided into three major subclasses (sigma, zeta, gamma) based on transcriptional analyses, one of which generates non-dividing

progenitors/progeny for the epithelium (zeta) and another postulated to be an intestinal-restricted subclass (gamma) (*van Wolfswinkel et al., 2014*). As neoblasts are the only mitotic cell type in planarians, they can be selectively ablated by irradiation, and over time, the immediate progeny of neoblasts eventually disappear as well (*Reddien et al., 2005b*; *Eisenhoffer et al., 2008*). Using fluorescence-activated cell sorting (FACS) analysis of irradiated animals stained with Hoechst, two cell populations can be discerned that are lost compared to non-irradiated animals: the 'X1' gate, which represents cells in the cell cycle with >2C DNA; and the 'X2' gate, which registers as a <2C DNA population due to Hoechst efflux. The remaining irradiation-insensitive (Xins) cells are assumed to be postmitotic differentiated cell types possessing 2C DNA (*Reddien et al., 2005b*; *Hayashi et al., 2006*). Studies have shown that the X1 population is >90% *piwi-1*[+], and this gate has been used as the stem cell fraction in comparative transcriptomic studies (*Reddien et al., 2005b*; *Eisenhoffer et al., 2008*; *Abril et al., 2010*; *Sandmann et al., 2011*; *Labbe et al., 2012*; *Onal et al., 2012*; *Resch et al., 2012*; *Solana et al., 2012*). The X2 cell population is only 10–20% *piwi-1*[+] and is thought to be enriched with immediate postmitotic progeny of neoblasts. This notion is supported by the finding that the five known markers of postmitotic progeny (*prog-1*, *prog-2*, and *AGAT-1/2/3*), predicted to label epithelial progenitors, are most highly expressed in the X2 cell fraction based on RNA-deep sequencing (RNAseq) (*Eisenhoffer et al., 2008*; *Labbe et al., 2012*; *Onal et al., 2012*; *van Wolfswinkel et al., 2014*). Additionally, other markers that associate with putative committed progenitor cells of the gut, brain, and eyes are also enriched in this cell fraction (*hnf4*, *chat*, and *sp6-9*, respectively) (*Wagner et al., 2011*; *Lapan and Reddien, 2012*; *Cowles et al., 2013*; *Scimone et al., 2014*). No study has comprehensively investigated the cells and transcripts specific to the X2 cell fraction, and the cell types within it remain an enigma.

Here, we hypothesized that regulators that drive stem cells toward postmitotic fates will be highly enriched in the stem cell progeny-associated X2 FACS fraction. Thus, we selected the top 100 transcripts enriched in this FACS gate, as well as 20 that were irradiation-sensitive with no X1 or X2 enrichment, to explore as putative markers or regulators. We found that while X2-enriched genes represented a heterogeneous mixture of cell types, transcripts expressed in epithelial progenitors comprised the predominant gene signature in the X2 fraction. We identified 32 new progeny markers and demonstrated that they are expressed predominantly in either *prog-1/2*[+] or *AGAT-1*[+] epithelial progenitors. In addition, *prog-1* and *prog-2* represent members of a larger gene family of unknown function, expressed throughout this epithelial lineage. Through RNA interference (RNAi) screening of the 120 candidate transcripts, we identified a homolog to the RNA-binding protein MEX3 (*Smed-mex3-1*) as a critical regulator of postmitotic stem cell progeny. Knockdown of *mex3-1* completely abolishes regenerative ability and halts the production of *prog-1/2*[+] and *AGAT-1*[+] postmitotic progeny populations. *piwi-1*[+] stem cells concomitantly increase in number, an increase that was observed in all three neoblast subclasses. Finally, *mex3-1(RNAi)* worms have impaired contribution to tissue turnover, as evidenced by drastically reduced production of lineage-restricted neoblast descendants and diminished cell addition towards multiple tissue types, in addition to the epithelium. These results suggest that *mex3-1* functions to maintain asymmetry in stem cell lineage progression by promoting postmitotic fates and suppressing self-renewal. Due to the well-known function of MEX3 in mediating asymmetric cell fates during *Caenorhabditis elegans* embryogenesis (*Draper et al., 1996*), we propose that *Smed-mex3-1* mediates a similar process in planarian stem cell lineages.

## Results

### RNAseq analysis of the progeny-associated X2 FACS cell fraction

Previously, we published two replicates of Illumina RNAseq of the X2 cell fraction to a depth of 206 million reads (*Labbe et al., 2012*). Here, we sequenced a third replicate to 63 million reads. We found a very high correlation across all of our sequencing replicates, as well as with two irradiated samples from a previous study, which we subsequently analyzed along with our irradiated sequencing (*Figure 1—figure supplement 1*, *Supplementary file 1*) (*Onal et al., 2012*; *Resch et al., 2012*; *Solana et al., 2012*). To identify transcripts enriched in the X2 cell fraction, we used the program DESeq (*Anders and Huber, 2010*) to compare RNAseq from purified X1 and X2 cells vs whole irradiated animals at 7 days after exposure to 60–100 Gray (Gy) of γ-irradiation (*Anders and Huber, 2010*; *Solana et al., 2012*; *Fernandes et al., 2014*). This identified 2839 X1 and 1512 X2 transcripts with a p-value ≤ 0.01 (*Figure 1A,B*, *Supplementary file 1*). It is important to note that X1 and X2 cells

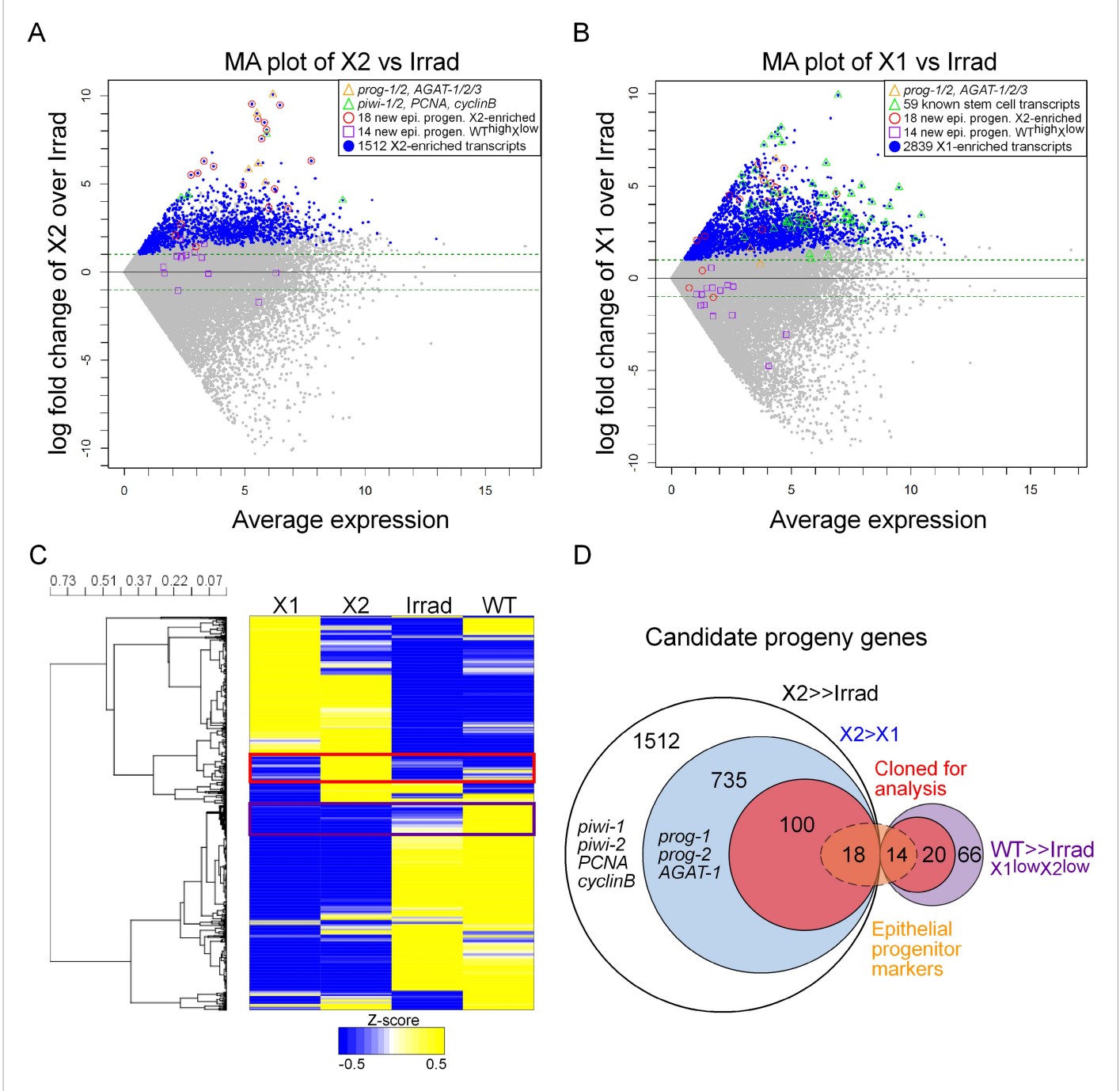

**Figure 1**. Transcriptional analysis of irradiation-sensitive cell populations in *Schmidtea mediterranea*. Irradiation-sensitive cell populations (X1, X2) isolated by fluorescence-activated cell sorting (FACS), lethally irradiated whole worms (Irrad), and intact control worms (WT) were analyzed by RNA-deep sequencing (RNAseq). (**A** and **B**) MA plots comparing enrichment in the X2 (**A**) or X1 (**B**) cell populations over Irrad, with average expression level. Each gray dot represents one transcript. Blue dots represent transcripts found to be significantly enriched in the respective cell population through DESeq analysis (X2, p < 0.01; X1, p < 0.001). Previously established postmitotic lineage markers are indicated by orange triangles. Previously established stem cell-specific transcripts are indicated as green triangles. Candidate genes identified in this study as markers of epithelial progenitors are indicated as red circles (X2-enriched) or purple boxes (WT$^{high}$X$^{low}$). (**C**) Hierarchical clustering of RNAseq data identifies transcripts enriched in the X2 population (red box), as well as a group of irradiation-sensitive transcripts, which have high expression in intact control worms but low expression in X1, X2, and Irrad populations (purple box, WT$^{high}$X$^{low}$). To improve visualization, the heatmap depicts z-scores scaled to the range of −0.5 to +0.5. (**D**) Selection of candidate progeny genes for analysis was determined by enriched expression in X2 fraction compared to both Irrad and X1 fractions. The pool of candidate genes validated in this study to be expressed in epithelial progenitors is indicated in orange. >> denotes more than fivefold enrichment.

*Figure 1. continued on next page*

*Figure 1. Continued*

The following figure supplements are available for figure 1:

**Figure supplement 1**. Statistical correlations between new and published data sets.

**Figure supplement 2**. Predicted protein alignments of the PROG family that are irradiation-sensitive.

shared the majority of their transcriptional profiles (*Figure 1C*), and *bona fide* progeny markers can be highly expressed in the X1 fraction, while *bona fide* stem cell markers can be highly expressed in the X2 fraction (*Figure 1A,B*). Therefore, to be considered X2-enriched, we imposed the additional criterion that the expression ratio of X2/X1 was >1, to exclude transcripts jointly expressed in both irradiation-sensitive populations. This eliminated previously known stem cell genes, such as *piwi-1* and *-2*, *PCNA* (proliferating cell nuclear antigen), and *cyclinB* (*Figure 1A*; 8th, 45th, 57th, and 80th highest enriched X2 genes, respectively), and reduced the total number of enriched X2 genes to 735 (*Figure 1D*, *Supplementary file 1*) (*Orii et al., 2005*; *Reddien et al., 2005b*; *Eisenhoffer et al., 2008*). Finally, we observed 66 transcripts that were highly expressed in wild-type animals, yet exhibited low counts in irradiated worms and in both X1 and X2 cell fractions (WT$^{high}$X$^{low}$, *Figure 1D*, *Supplementary file 1*). From the remaining 735 X2-specific genes as well as these 66 other irradiation-sensitive transcripts, we hypothesized that these represented multiple types of irradiation-sensitive progenitor cells. We next cloned the top 100 X2-specific and 20 WT$^{high}$X$^{low}$ transcripts for expression and functional analyses (*Figure 1D*, *Supplementary file 2*). Genes were annotated based on the top BLAST hit in mouse, when the Expect value passed the threshold of $e^{-5}$.

Interestingly, the predicted proteins encoded by *prog-1* and *prog-2* do not have clear homology to genes in other animals, including the genomes of other sequenced flatworms (*Echinococcus multilocularis*, *Schistosoma mansoni*, *Macrostomum lignano*—www.macgenome.org) (*Berriman et al., 2009*; *Zheng et al., 2013*). However, we observed that they had low similarity to each other and represented a family of at least 24 distinct members across multiple transcriptomes in *S. mediterranea*, 15 of which were represented in the top 100 X2-enriched gene set (*Sandmann et al., 2011*; *Solana et al., 2012*; *Currie and Pearson, 2013*; *Vogg et al., 2014*). Translations of the predicted open reading frames (average size 179 amino acids) for these 15 *prog*-related genes were aligned and analyzed by protein domain prediction software SMART (*Figure 1—figure supplement 2*) (*Schultz et al., 1998*; *Letunic et al., 2014*). The only motif that could be detected was a signal sequence at the N-terminal end of the predicted proteins, suggesting that these proteins are secreted. These PROG-1/2 homologous genes were then named in a numbered sequence based on their closest homolog (e.g., *prog-1-1*, *prog-2-1*).

## Whole-mount expression analysis of X2-enriched and WT$^{high}$X$^{low}$ transcripts reveals 32 markers of epithelial progenitors

To begin our analysis of the top 100 X2-enriched transcripts and the top 20 WT$^{high}$X$^{low}$ transcripts, we performed whole-mount in situ hybridization (WISH) to elucidate gene expression patterns. We observed that 40/120 transcripts were either not detectable or not specific, and the remaining 80/120 genes could be binned into one of four categories (*Figure 2A*, *Figure 2—figure supplement 1*). One category (13/120) contained genes with the most intense expression in the bi-lobed brain and nervous system, with low levels of expression elsewhere in the body. A second group of genes (18/120) exhibited a predominantly stem cell-like expression pattern, with or without brain expression, which was confirmed by high expression in the X1 cell fraction (*Figure 2A*, *Figure 2—figure supplement 1A*, *Supplementary file 2*). A third subset of genes was expressed in a variety of distinct patterns including the gut, pharynx, peri-pharyngeal region, and neck (17/120) (*Figure 2A*, *Figure 2—figure supplement 1A*). Finally, the fourth and largest group (32/120) consisted of genes with an expression pattern highly similar to those of the known early and late progeny markers *prog-1*, *prog-2*, and *AGAT-1*, which are sub-epithelial across the entire animal with expression anterior to the photoreceptors (*Figure 2A*, *Figure 2—figure supplement 2*) (*Eisenhoffer et al., 2008*). This *prog*-like subset exhibited varying degrees of X2-enrichment, and some were highly expressed in the X1 population (*Figure 1A*, *Supplementary file 2*). Transcripts that displayed a *prog*-like pattern where

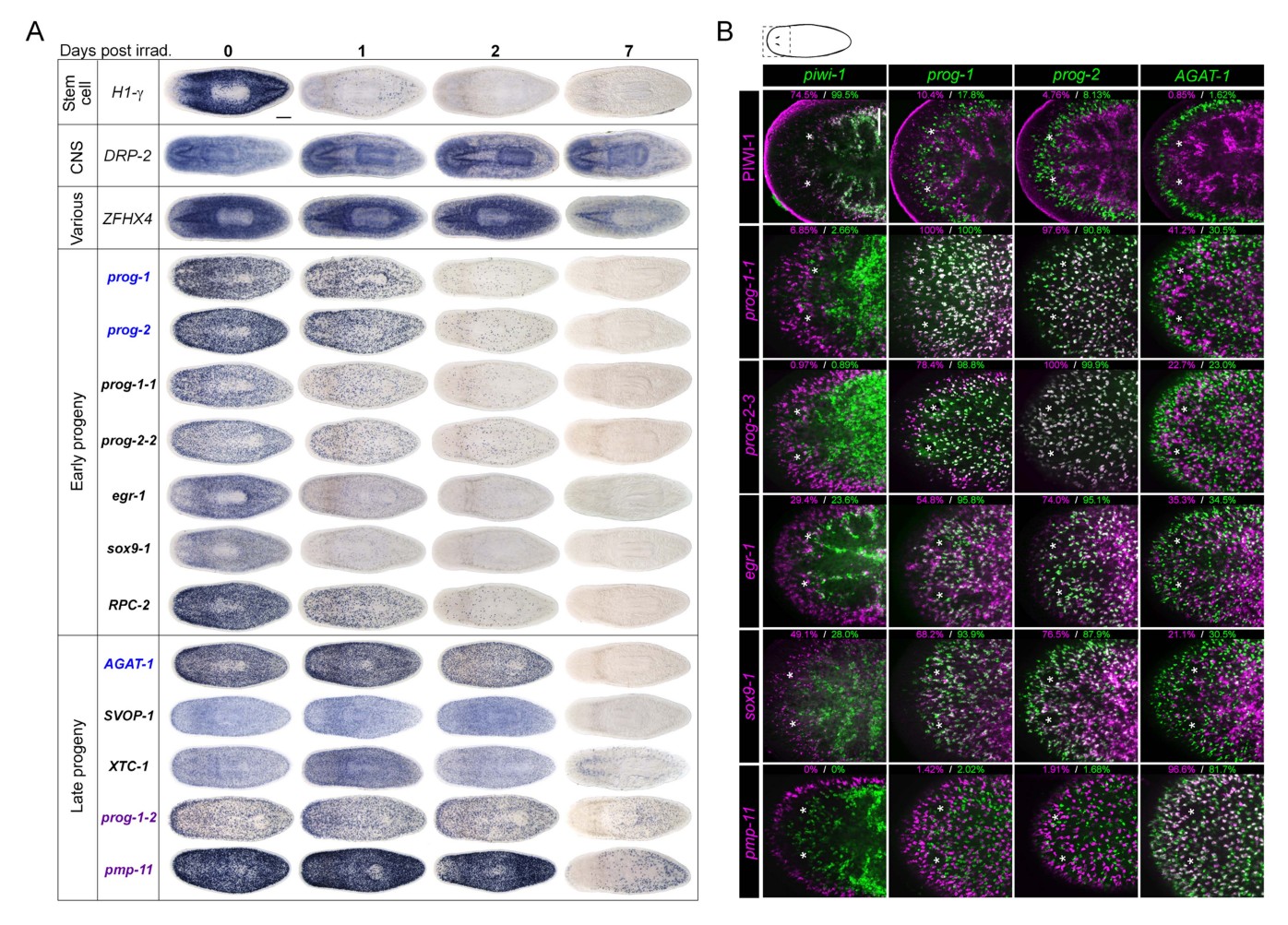

**Figure 2.** Expression analyses of candidate progeny genes. (**A**) Whole-mount in situ hybridization (WISH) analysis of X2-enriched and WT$^{high}$X$^{low}$ candidate progeny genes in control and lethally irradiated worms. Examples of genes expressed in a stem cell-like pattern, strong in the bi-lobed brain, distinct and unique patterns, or in a *prog*-like sub-epithelial pattern are shown. *prog*-like genes were categorized as early progeny markers when they displayed similar post-irradiation down-regulation kinetics as *prog-1* and *prog-2*, or late progeny markers when they displayed similar loss kinetics as *AGAT-1*. Established lineage markers *prog-1*, *prog-2*, and *AGAT-1* are included as comparisons and highlighted in blue text. Candidate epithelial progenitor genes with WT$^{high}$X$^{low}$ expression are indicated in purple text. Anterior, left; scale bar, 200 μm. (**B**) Combinatorial double fluorescent WISH (dFISH) between lineage markers and identified X2-enriched and WT$^{high}$X$^{low}$ postmitotic progeny markers was performed to assess co-localization with stem cell, early progeny, and late progeny cell populations. Percent co-localization is shown at the top of each panel and is averaged from 3 to 4 animals. *Magenta*, percent co-expression in [*row gene*]$^+$ cells; *green*, percent co-expression in [*column gene*]$^+$ cells. Images from the head, trunk, and tail regions were used for all cell counts, with a minimum of 300 cells counted. Dispersions of data are in **Supplementary file 3**. Representative confocal projections spanning 4–6 μm of the head region are shown. Eyespots are marked by asterisks. Anterior, left; scale bar, 100 μm.

The following figure supplements are available for figure 2:

**Figure supplement 1**. Gene expression analysis of candidate progeny genes.

**Figure supplement 2**. Gene expression analysis of candidate progeny genes with a *prog*-like expression pattern.

**Figure supplement 3**. Co-localization of progeny markers with dFISH.

**Figure supplement 4**. Analysis of new progeny markers in *zfp-1(RNAi)* animals.

BLAST did not identify significant similarity were named as *postmitotic progeny* (*pmp*'s) with ascending numerals (e.g., *pmp-3*, *pmp-4*).

Using WISH, we confirmed that transcripts from each of these expression categories were irradiation-sensitive (*Figure 2A*, *Figure 2—figure supplements 1B, 2*). Genes with a stem cell-like component to the expression pattern exhibited down-regulation after 24 hr, as anticipated for *bona fide* stem cell markers. However, we observed that multiple genes with distinct tissue expression patterns did not change appreciably following irradiation (*Figure 2—figure supplement 1B*) and were deemed unlikely to be candidates of progenitor cell types. We subsequently focused on the group of genes exhibiting an epithelial progenitor-like pattern, given their prevalence, and sought to determine whether they were co-expressed in previously described *prog-1*$^+$ and *AGAT-1*$^+$ populations or specified towards other unknown lineages.

The postmitotic progeny markers *prog-1/2* and *AGAT-1/2/3* not only have largely non-overlapping expression patterns, but the kinetics of down-regulation following irradiation substantially differ as well (*Figure 2A*) (*Eisenhoffer et al., 2008*). *prog-1/2* expression is lost within 48 hr post-irradiation, while *AGAT-1* expression is lost between 48 hr and 7 days, leading to these two cell populations being termed 'early' and 'late' progeny, respectively. Combined with evidence from BrdU-labeling studies, the early and late progeny populations have been proposed to reflect a spatiotemporal progression along one lineage of ASC differentiation destined for the epithelium, such that *prog-1/2* expression represents the transition state between neoblasts (potentially zeta-neoblasts) and *AGAT-1*$^+$ cells (*Eisenhoffer et al., 2008*; *Pearson and Sánchez Alvarado, 2010*; *van Wolfswinkel et al., 2014*). To demonstrate that our newly identified *prog*-like markers were irradiation-sensitive and to determine their kinetics of down-regulation, animals were lethally irradiated (60 Gy) and analyzed for each marker for 7 days. We observed that 13/32 transcripts showed similar kinetics of down-regulation as the early progeny markers *prog-1* and *prog-2* and were almost completely lost by 48 hr post-irradiation (*Figure 2A*, *Figure 2—figure supplement 2*). The remaining 19/32 transcripts showed the majority of loss beyond 48 hr post-irradiation, consistent with being late progeny markers (*Figure 2A*, *Figure 2—figure supplement 2*). Interestingly, all WT$^{high}$X$^{low}$ genes grouped as late progeny, and moreover, were the only genes with detectable expression at 7 days post-irradiation. These highly similar expression patterns and irradiation-sensitivity kinetics were strong indicators that these new progeny markers were also expressed in epithelial progenitors. Through co-localization analyses, we next investigated whether that was the case, or whether these markers represented novel irradiation-sensitive stem cell progeny.

## New progeny markers label either early or late progeny of the epithelial lineage

Prior to determining how the new progeny markers fit into this putative lineage using double fluorescent WISH (dFISH), we first duplicated previous experiments with *piwi-1*, *prog-1*, and *AGAT-1* in order to establish baseline values of overlap. Congruent with previous studies, we found a small amount of overlap between *piwi-1* with *prog-1*, and minimal with *AGAT-1*: 7.2% ± 2.1 of *prog-1*$^+$ cells were *piwi-1*$^+$, while merely 0.29% ± 0.50 of *AGAT-1*$^+$ cells were *piwi-1*$^+$ (*Supplementary file 3*). In contrast to a previous study that found nearly 45% overlap between the progeny populations (*Eisenhoffer et al., 2008*), we found that only 5.4% ± 2.4 of *prog-1*$^+$ cells co-expressed low levels of *AGAT-1*, while 5.2% ± 3.2 of *AGAT-1*$^+$ cells had *prog-1* expression. dFISH with new progeny markers also showed little overlap between early and late progeny, confirming that these progeny transcripts mark two mainly non-overlapping populations (*Figure 2B*, *Figure 2—figure supplement 3*), and that previous overlap was likely an dFISH artifact. It has been shown that PIWI-1 protein has a wider expression domain than *piwi-1* mRNA, reflecting the perdurance of PIWI-1 into postmitotic progeny (*Guo et al., 2006*; *Wenemoser and Reddien, 2010*). It is unknown how differentiated these *piwi-1*$^-$PIWI-1$^+$ cells are, but they are clearly in a transition from a stem cell gene expression state to various postmitotic fates. In support of these data, we found that 17.8% ± 12.2 of *prog-1*$^+$ cells to be PIWI-1$^+$, while 1.6% ± 1.2 of *AGAT-1*$^+$ cells had detectable PIWI-1 expression (*Figure 2B*, *Supplementary file 3*).

To determine whether our newly identified early and late progeny markers represented distinct population(s) of descendent cells outside of the putative *prog-1/prog-2/AGAT-1* lineage, pairwise dFISH was performed and quantified. Consistent with the kinetics of loss post-irradiation, dFISH

uncovered extensive overlap between the new early progeny markers with *prog-1* and *prog-2* (*Figure 2B*, *Figure 2—figure supplement 3*, *Supplementary file 3*). Some genes, such as *prog-1-1* and *prog-2-3*, exhibited nearly 100% co-localization with *prog-1* and *prog-2*, respectively, whereas the lowest percentage overlap was observed with *egr-1* and *sox9-1*, with approximately 50–70% of these cells co-expressing *prog-1*. *sox9-1* and *egr-1* were previously identified to be expressed in zeta-class *piwi-1*+ neoblasts (referred to as *soxP-3* and *egr-1* [*Wagner et al., 2012*]), and we confirmed that a sizeable percentage of *sox9-1*+ and *egr-1*+ cells co-expressed *piwi-1* (49.1% ± 5.2 and 29.4% ± 11.9, respectively, *Supplementary file 3*). Almost all newly identified late progeny transcripts examined were found to exhibit substantial co-expression with *AGAT-1*, with overlap ranging from 100% with *prog-1-7* to 55.5% with *prog-1-4* (*Figure 2—figure supplement 3*, *Supplementary file 3*). The exception was the late progeny marker *prog-1-2*, where only 20.4% of *prog-1-2*+ cells co-expressed *AGAT-1* (*Figure 2—figure supplement 3*). Given that *prog-1-2* is expressed in both subepithelial and epithelial cells, we propose that *prog-1-2*+ cells represent a more differentiated state along the epithelial lineage and are possibly the subsequent transition for *AGAT-1*+ cells. We also performed dFISH between newly identified early and late progeny markers, which similarly showed that genes within each category exhibit highly overlapping expression (*Figure 2—figure supplement 3*, *Supplementary file 3*).

Finally, to validate that expression of these genes marked epithelial progenitors, we examined their expression by WISH in *zfp-1(RNAi)* worms. Knockdown of *zfp-1* has been demonstrated to specifically ablate epithelial progenitors and epithelial differentiation, but not the differentiation of other tissue types (*van Wolfswinkel et al., 2014*). We observed that every new epithelial progenitor marker assessed was down-regulated after *zfp-1* RNAi, confirming that these genes were indeed expressed in epithelial progenitors (*Figure 2—figure supplement 4*). Together, these findings demonstrated that our newly identified transcripts represent markers of the early and late progeny produced by zeta neoblasts and comprise the primary two progenitor populations en route to the epithelium (*van Wolfswinkel et al., 2014*).

## RNAi screening identifies *mex3-1* as a candidate regulator of differentiation

The self-renewal of ASCs and the appropriate differentiation of postmitotic progeny are the driving force behind homeostatic cell turnover and regeneration of all tissues in planarians (*Newmark and Sánchez Alvarado, 2000*; *Rossi et al., 2008*; *Baguna, 2012*; *van Wolfswinkel et al., 2014*). RNAi against genes required for differentiation, such as *p53*, *CHD4*, *zfp-1*, and *vasa-1*, results in the decline of postmitotic progeny without depletion of ASCs, and subsequent defects in tissue homeostasis and regeneration (*Pearson and SánchezAlvarado, 2010*; *Scimone et al., 2010*; *Wagner et al., 2012*). To determine whether progeny-enriched genes were regulators of postmitotic fates, we used RNAi knockdown against our set of 100 X2-enriched and 20 WT[high]X[low] genes and screened for the above phenotypes. In agreement with previous data, RNAi against *prog-1* and *prog-2* separately or together did not yield any detectable phenotype (*Eisenhoffer et al., 2008*). Unexpectedly, none of the new epithelial progenitor markers yielded phenotypes upon knockdown either, suggestive of considerable functional redundancy among these genes. In contrast, knockdown of a gene encoding a homolog to the RNA-binding protein MEX3, *Smed-mex3-1* (*mex3-1*) produced phenotypes highly suggestive of defective stem cell lineage progression. *mex3-1(RNAi)* was completely penetrant and lethal, resulting in ventral curling, head regression, and dorsal lesioning during homeostasis, as well as loss of regenerative ability after amputation (*Figure 3A–C*), indicative of epithelial homeostasis defects as well as overall stem cell impairment (*Bardeen and Baetjer, 1904*; *Reddien et al., 2005a*). Using reciprocal BLAST, we identified two other MEX3 homologs in *S. mediterranea* (*mex3-2* and *mex3-3*, transcripts SmedASXL_000637 and SmedASXL_01505, respectively), which both contained two KH RNA-binding domains and had top reciprocal BLAST hits in mouse, fly, and *C. elegans*. These additional MEX3 homologs were neither irradiation-sensitive nor produced observable phenotypes after knockdown and thus not investigated further (*Figure 3—figure supplement 1*).

To begin elucidating the mechanism by which *mex3-1* potentially regulates stem cell lineage progression, we examined where the gene was expressed in the epithelial lineage. By RNAseq, *mex3-1* showed >fivefold enrichment in the X2 fraction over irradiated worms but also exhibited significant expression in the X1 stem cell fraction (*Figure 3D*). WISH of wild-type worms revealed a stem cell-like

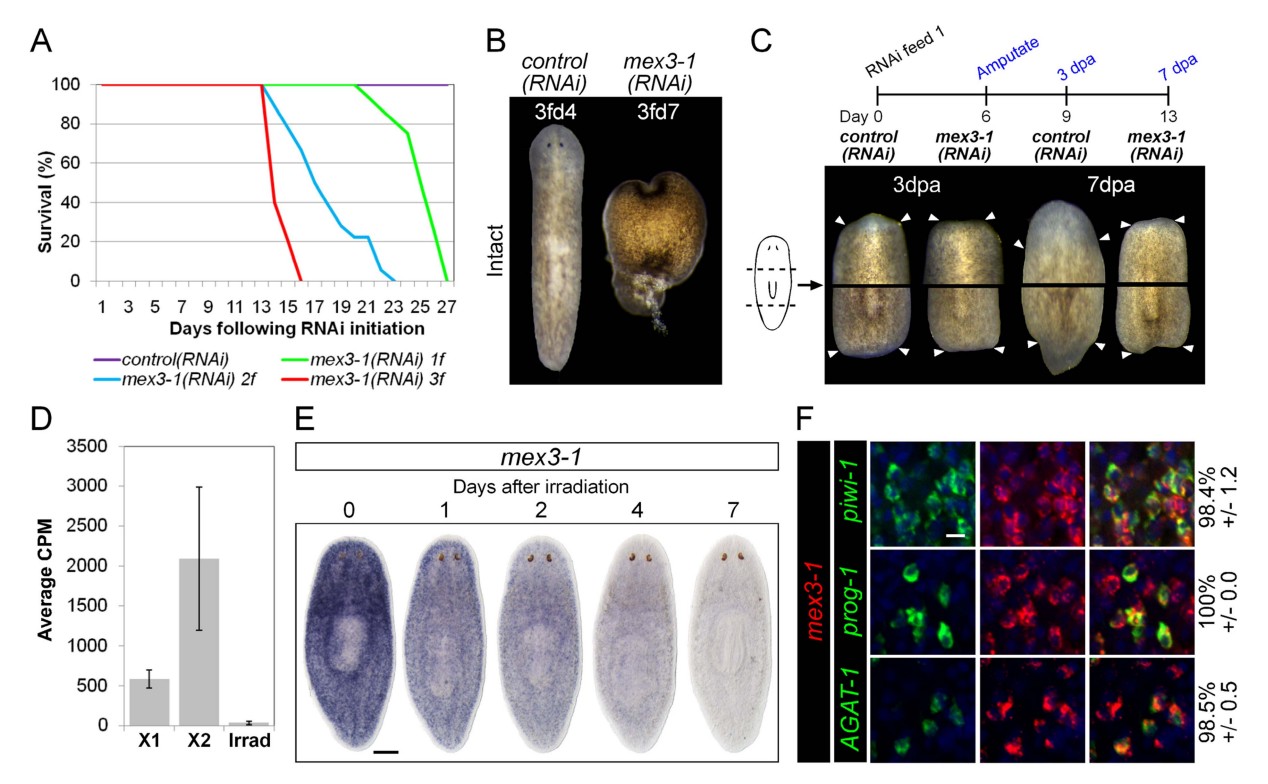

**Figure 3**. *mex3-1* is required for tissue homeostasis and regeneration. (**A**) Survival curves of *mex3-1(RNAi)* animals to determine the optimal RNAi dosage. One RNAi feed was sufficient to produce completely penetrant lethality by 27 days. (**B**) RNAi worms were observed for homeostatic abnormalities after 1–3 feeds. Time point at which animals were imaged is indicated as *x* days after *z* feeds (*zfdx*). (**C**) Regenerative ability after injury was tested according to the experimental timeline shown. Trunk fragments after pre- and post-pharyngeal amputations are shown. (**D**) Expression levels of *mex3-1* in different FACS populations by RNAseq. Error bars show standard deviation. (**E**) WISH analysis of *mex3-1* in intact worms after 60 Gray (Gy) irradiation. Scale bar, 200 μm. (**F**) dFISH was performed to examine *mex3-1* expression in stem cells and postmitotic progeny. Numbers indicate the percentage of stem cells, early progeny, or late progeny co-expressing *mex3-1* (n > 400 cells per dFISH, ± standard error). Scale bar, 10 μm.

The following figure supplement is available for figure 3:

**Figure supplement 1**. Identification and analysis of MEX3 homologs in *S. mediterranea*.

expression pattern with substantial expression peripheral to the stem cell compartment (*Figure 3E*). Following irradiation, progressively more of the pattern was lost over 7 days (*Figure 3E*), consistent with expression in both stem cell and immediate postmitotic progeny populations, and also consistent with previous irradiation data for *mex3-1* (*Solana et al., 2012*). Analysis of *mex3-1* using dFISH with lineage markers confirmed that *mex3-1* was widely expressed in stem cells, early progeny, and late progeny (*Figure 3F*). Therefore, the homeostasis and regeneration defects described above could result from defects in any one or all of these cell populations, which was subsequently tested.

## *mex3-1* is required for epithelial progenitor specification

To ascertain which step(s) in stem cell lineage progression were aberrant after *mex3-1* knockdown, we performed WISH analysis to follow stem cell and postmitotic *prog-1/2+* and *AGAT-1+* progeny population dynamics after RNAi was initiated. Knockdown of *mex3-1* leads to a rapid decline of the two progeny populations but not of stem cells (*Figure 4A*, *Figure 4—figure supplement 1A*), with the majority of progeny gene expression lost by 6 days after RNAi (*Figure 4—figure supplement 1B*). We assessed the expression of additional newly identified epithelial progenitor markers as well and observed that all were similarly abolished, supporting the loss of these two progeny types (*Figure 4B*).

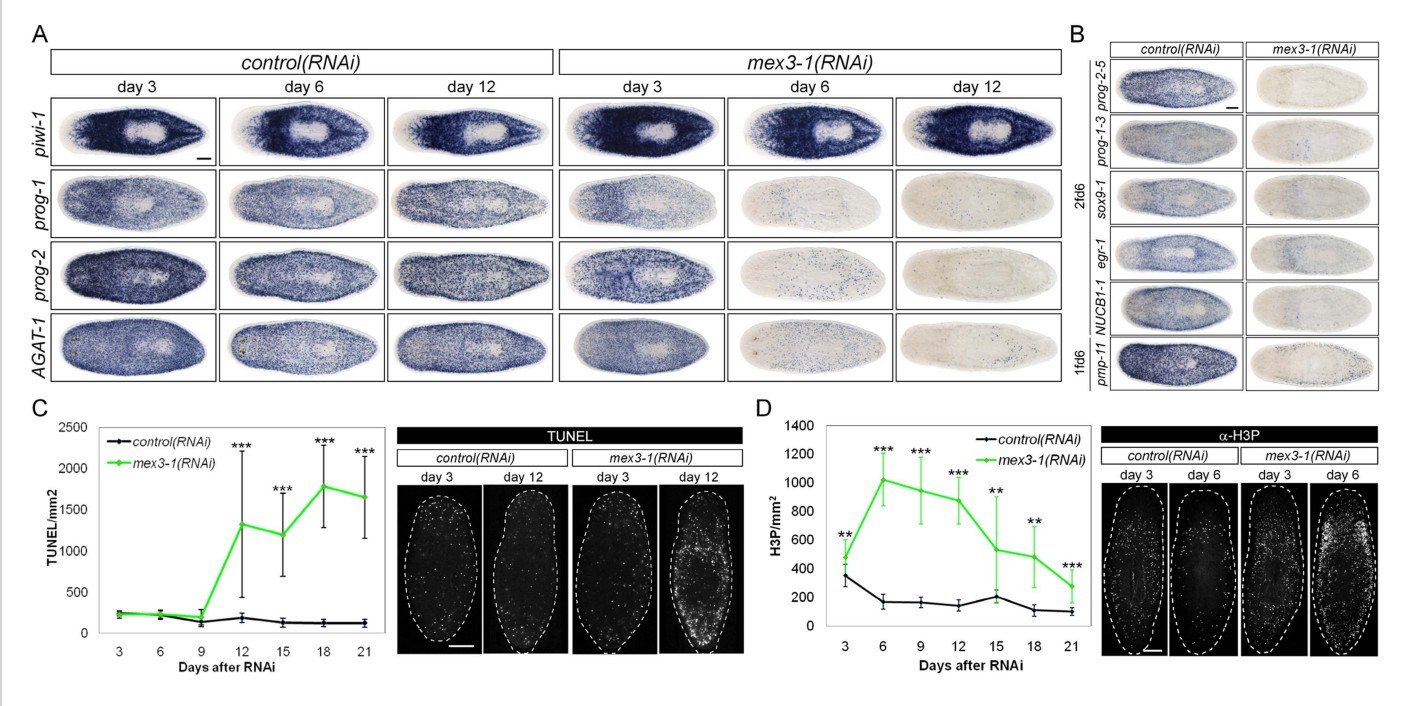

**Figure 4**. RNAi against *mex3-1* selectively affects progeny markers and causes hyper-proliferation. (**A**) Lineage markers labeling stem cells (*piwi-1*), early (*prog-1, prog-2*), and late (*AGAT-1*) progeny were assessed by WISH after RNAi. The day 12 time point reflects the maximal perturbation to progeny markers prior to obvious health decline in the animal. (**B**) Representative newly identified early and late progeny transcripts were assessed by WISH after RNAi. (**C**) Whole-animal quantification of TUNEL was performed to measure cell death in RNAi animals. Representative stains and time points are shown to the right. (**D**) Whole-animal quantification of H3P immunolabeling was performed to assess cell proliferation after RNAi. Representative stains and time points are shown to the right. Unless otherwise noted, all experimental time points are indicated as after a single RNAi feeding. Scale bars, 200 μm. Error bars are standard deviations. **p < 0.01, ***p < 0.001 (Student's *t*-test).

The following figure supplement is available for figure 4:

**Figure supplement 1**. *mex3-1* RNAi depletes progeny without impairing stem cell proliferation.

The down-regulation of progeny gene expression without corresponding decreases in the stem cell population suggested a selective defect in specifying committed progeny. However, these results could similarly arise from defective maintenance and survival of progeny, thus, we examined levels of apoptosis with TUNEL (terminal deoxynucleotidyl transferase dUTP nick end labeling) during phenotypic progression. Quantification of TUNEL showed that *mex3-1(RNAi)* animals had comparable levels of cell death to control worms up until 9 days after RNAi, but significantly higher levels from 12 days and onward (*Figure 4C*). Given that progeny gene expression was mostly ablated by 6 days after RNAi, we concluded that the loss of progeny populations was not attributed to progeny cell death. The cause of the late rise in cell death remains unclear but is coincident with when morphological homeostatic phenotypes begin to manifest and may be a secondary effect of declining animal health. We next investigated whether impaired stem cell proliferation was the underlying cause of reduced progeny production and performed time-course analysis of phosphorylated histone H3 immunolabeling (H3P). We observed significantly increased levels of proliferation in *mex3-1(RNAi)* worms at every time point examined (*Figure 4D*), which suggested that stem cell division was not impeded by *mex3-1* knockdown.

To exclude the possibility that the heightened H3P⁺ levels in *mex3-1(RNAi)* worms reflected an accumulation of cells arrested in G2/M phase and an inability to complete the cell cycle, we performed labeling with the thymidine analog F-*ara*-EdU to measure S-phase progression. Worms pulsed at either 6 or 9 days after RNAi and fixed 24 hr later showed significantly increased numbers of total EdU-labeled cells in *mex3-1(RNAi)* animals compared to controls (*Figure 5A*), demonstrating that during phenotypic progression, a greater number of cells were entering the cell cycle. FACS profiles

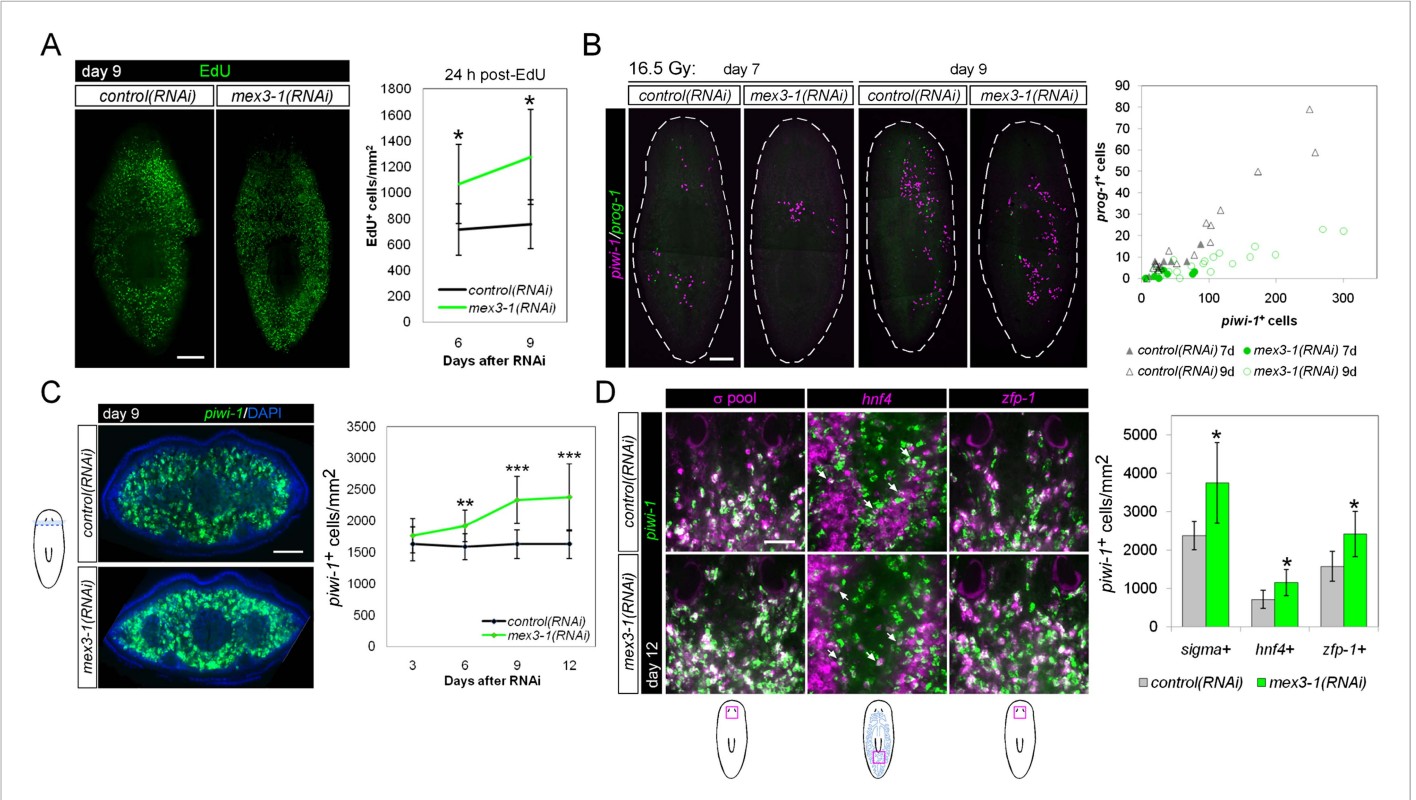

**Figure 5**. *mex3-1(RNAi)* animals exhibit expansion of the stem cell compartment. (**A**) RNAi worms were administered EdU at 6 or 9 days after RNAi and quantified after a 24-hr period. Counts were performed on whole-animal single confocal planes. (**B**) *mex3-1(RNAi)* animals were irradiated with a sublethal dose (16.5 Gy), and stem cells (*piwi-1*) and progeny (*prog-1*) were quantified by dFISH at 7 and 9 days after irradiation. The proportion of progeny to stem cells between *mex3-1(RNAi)* and control worms differed significantly ($p < 0.001$, analysis of covariance). Whole-animal confocal projections are shown. Each point on the graph represents one animal. (**C**) Stem cells were quantified in pre-pharyngeal cross sections of intact worms after RNAi by *piwi-1* fluorescent WISH (FISH) during phenotypic progression. Single confocal planes at day 9 after RNAi are shown; dorsal, top. (**D**) Quantification of stem cell subclasses in intact worms 12 days after RNAi. Stem cell subclass was determined by *piwi-1* labeling and expression of *soxP-1* pooled with *soxP-2* (sigma subclass), *hnf4* (gamma), or *zfp-1* (zeta). Diagrams indicate areas of worms quantified, and arrows indicate example double-positive cells. Scale bars, 200 μm in (**A**–**C**) and 50 μm in (**D**). Error bars represent standard deviation. *$p < 0.05$, **$p < 0.01$, ***$p < 0.001$ (Student's *t*-test).

The following figure supplement is available for figure 5:

**Figure supplement 1**. Quantification of stem cells in *mex3-1(RNAi)* animals.

of *mex3-1(RNAi)* animals at day 9 revealed relatively normal proportions of cells in both the X1 and X2 gates (*Figure 4—figure supplement 1C*), further evidence that stem cells were progressing through the cell cycle and not arrested at 4C DNA in the X1 gate. Worms exposed to sublethal doses of irradiation initially lose the vast majority of their stem cells and immediate postmitotic descendants; over time, the stem cell and progeny populations gradually recover, offering an easily quantifiable approach to assess both self-renewal and differentiation (*Wagner et al., 2012*). Sublethally irradiated *mex3-1(RNAi)* worms expanded *piwi-1*[+] stem cell numbers over time but produced disproportionately fewer *prog-1*[+] progeny than control worms (*Figure 5B*). These data demonstrated that the loss of early and late progeny marker expression after *mex3-1* knockdown could not be attributed to cell cycle arrest or increased cell death, but rather result from a failure in cell fate specification.

## Knockdown of *mex3-1* results in expansion of the stem cell compartment

Given that early and late progeny cell fates failed to be specified and adopted in *mex3-1(RNAi)* animals despite ongoing stem cell divisions, we hypothesized that there was a loss in stem cell lineage asymmetry in favor of stem cell self-renewal. We sought to determine whether this was the case by

quantifying *piwi-1*[+] stem cells in transverse cross sections after RNAi. By 6 days after RNAi, we observed highly significant increases in the number of *piwi-1*[+] stem cells in *mex3-1(RNAi)* animals compared to controls, which rose to a 50% increase by day 12 (*Figure 5C*, *Figure 5—figure supplement 1*). The expansion of *piwi-1*[+] stem cells after *mex3-1* knockdown could either reflect a global increase in all stem cell subclasses or reflect an increase in a specific subclass (zeta-, sigma-, and gamma-neoblasts) (*van Wolfswinkel et al., 2014*). To ascertain whether subclasses were selectively affected, we quantified the number of *piwi-1*[+] stem cells belonging to each subclass using the following probes: *zfp-1* for the zeta subclass, *hnf4* for the gamma subclass, and a pooled mix of *soxP-1* and *soxP-2* for the sigma subclass. Assessment at 12 days after RNAi revealed that all three subclasses were significantly increased by approximately 50% in *mex3-1(RNAi)* animals compared to controls (*Figure 5D*). Together, these results demonstrated that the failure to specify epithelial progenitors in *mex3-1(RNAi)* animals was not due to loss of zeta-neoblasts, and suggested that the expansion of the stem cell pools was due to an imbalance in cell fates favoring stem cell self-renewal over differentiation.

## mex3-1 is required for turnover of multiple tissues

The ventral curling defect during homeostasis and ablation of all early and late progeny markers in *mex3-1(RNAi)* worms suggested that epithelial turnover may be severely compromised. We performed EdU labeling in intact *mex3-1(RNAi)* animals to measure the entry of new cells into the epithelium under normal homeostatic conditions. 7 days following EdU administration, numerous EdU[+] cells were present in the epithelium of *control(RNAi)* worms, but virtually none had been incorporated into the epithelium in *mex3-1(RNAi)* animals (*Figure 6A*). Additionally, following amputation, *mex3-1(RNAi)* worms failed to re-establish the expression of epithelial genes at wounding sites, demonstrating that during both homeostasis and regeneration, *mex3-1* is required for specification and differentiation of epithelial cell types (*van Wolfswinkel et al., 2014*) (*Figure 6B*).

Previously, it was shown that even by selectively abolishing zeta-neoblasts and epithelial turnover, a regenerative blastema can still form with differentiated tissues such as brain, protonephridia, intestine, and muscle (*van Wolfswinkel et al., 2014*). Given that *mex3-1(RNAi)* animals were unable to produce any regenerative blastema, we hypothesized that *mex3-1* may have a crucial role in the broad specification of multiple lineages. To examine this possibility, we assessed *mex3-1(RNAi)* worms for changes in the number of lineage-specified neoblast progeny of other tissue types in various body regions (*Figure 6C*). As PIWI-1 protein persists in immediate postmitotic stem cell descendants for up to 72 hr (*Guo et al., 2006*), co-localization of PIWI-1 with tissue-specific markers was used to identify lineage-restricted neoblast descendants, encompassing undifferentiated progenitors and newly differentiating cells. The neural genes, *chat* and *coe*, eye-specific transcription factor *ovo*, pharyngeal marker *FoxA*, and protonephridial marker *six1/2-2* were used as markers to indicate differentiation toward their respective tissues (*Scimone et al., 2011*; *Wagner et al., 2011*; *Lapan and Reddien, 2012*; *Cowles et al., 2013*; *Adler et al., 2014*). Quantification of lineage-restricted neoblast progeny in intact *mex3-1(RNAi)* worms 12 days after RNAi showed significant reductions in all examined cell types (*Figure 6D*). We observed that virtually all *chat*[+]PIWI-1[+] cells expressed *mex3-1* (*Figure 6—figure supplement 1A*), suggesting a direct role for *mex3-1* in regulating differentiation outside the epithelial lineage. Concordant with decreased production of lineage-restricted neoblast progeny, we also observed that 5 days following labeling with BrdU, the entry of new cells into brain, intestine, and pharynx was significantly decreased in *mex3-1(RNAi)* animals (*Figure 6E*, *Figure 6—figure supplement 1B*). These data demonstrate that the diminished capacity of *mex3-1(RNAi)* animals to produce differentiated progeny is not restricted to the epidermal lineage but is characteristic of multiple lineages in planarians.

We examined whether impaired differentiation toward multiple tissues could be observed in *p53 (RNAi)* animals as well, as *p53* knockdown has previously been shown to deplete *prog-1*[+] progeny and increase stem cell proliferation (*Pearson and Sánchez Alvarado, 2010*). We found that knockdown of *p53* did not alter the numbers of PIWI-1[+]*ovo*[+] eye- or PIWI-1[+]*chat*[+] brain-specified neoblast progeny (*Figure 6—figure supplement 2*), demonstrating a broader role for *mex3-1* in differentiation. To determine whether *mex3-1* knockdown resulted in a global impediment in generating postmitotic

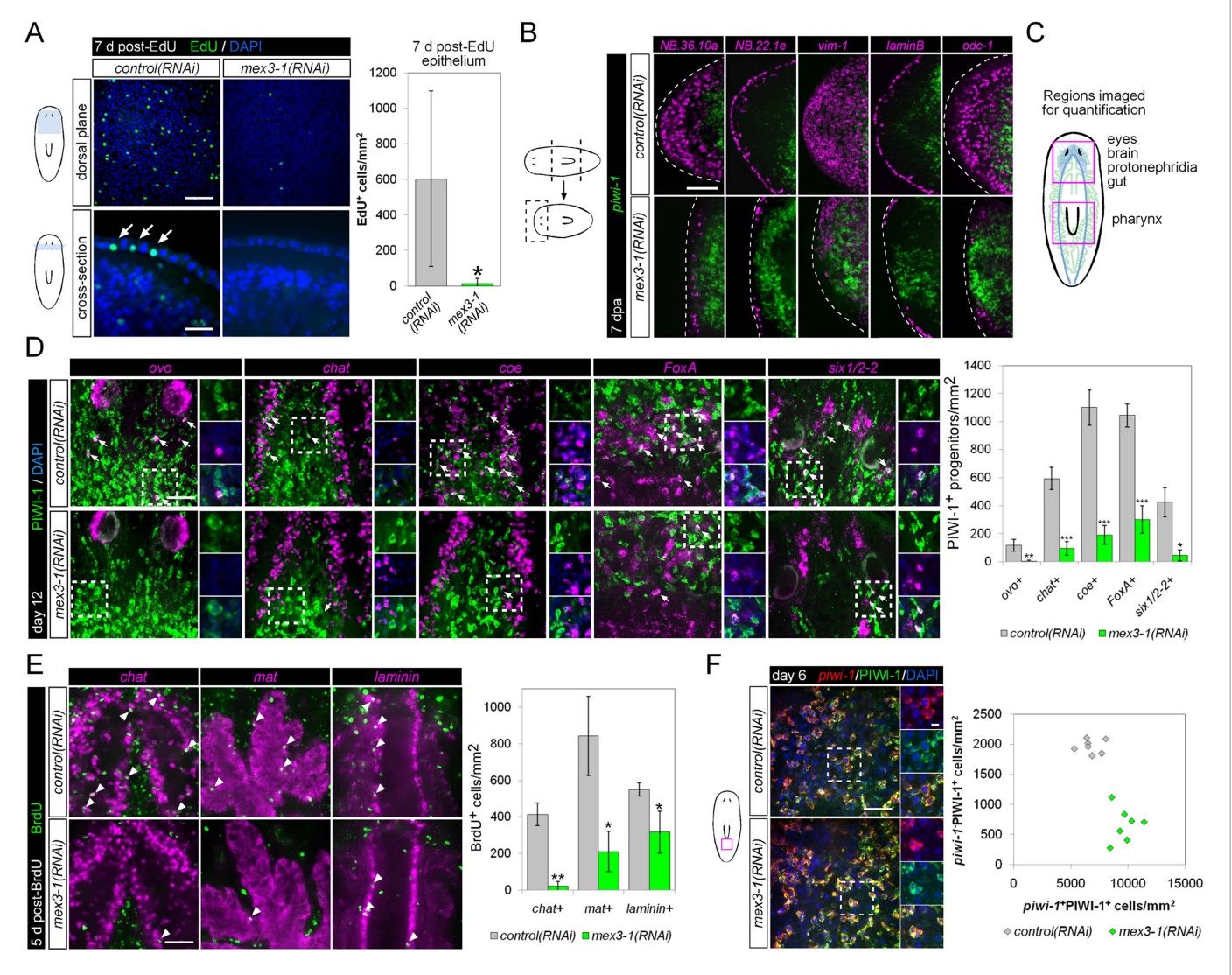

**Figure 6**. *mex3-1* is required for epithelial turnover and regeneration as well as differentiation toward multiple tissues. (**A**) RNAi animals were administered EdU 6 days after RNAi, and EdU$^+$ labeling in the epithelium was quantified after a 7-day period. Single confocal planes are shown, with EdU$^+$ cells in the epithelium indicated by arrows. Top panel scale bar, 100 μm; bottom panel scale bar, 50 μm. (**B**) RNAi animals amputated 7 days after RNAi were analyzed after 7 days regeneration by dFISH for stem cells and epithelial-associated genes. Single confocal planes or projections of head blastemas are shown, with anterior to the left. Dotted lines indicate animal boundary. Scale bar, 100 μm. (**C**) Diagram indicating regions of animal imaged and examined for quantification of lineage-restricted neoblast progeny. (**D**) Lineage-restricted progeny populations were quantified in intact worms 12 days after RNAi, using PIWI-1 immunolabeling and FISH for *ovo* (eyes), *chat* or *coe* (brain), *FoxA* (pharynx), and *six1/2-2* (protonephridia). Single confocal planes are shown. Arrows indicate example double-positive cells. Magnified areas are indicated by dashed boxes and inset to the right of each image. Scale bar, 50 μm. (**E**) RNAi animals were administered BrdU 6 days after RNAi, and BrdU$^+$ labeling in differentiated tissues (*chat*, brain; *mat*, gut; *laminin*, pharynx) was examined after a 5-day period. Single confocal planes are shown. Arrowheads indicate example double-positive cells. Scale bar, 50 μm. Error bars represent standard deviation. *p < 0.05, **p < 0.01, ***p < 0.001 (Student's *t*-test). (**F**) Quantification of stem cells (*piwi-1*$^+$PIWI-1$^+$) and immediate postmitotic stem cell descendants (*piwi-1*$^-$PIWI-1$^+$) were performed 6 days after RNAi, in head, pre-pharyngeal, and tail regions. Shown are single confocal planes from the tail region. Magnified areas are indicated by dashed boxes. The proportion of postmitotic descendants to stem cells between *mex3-1* *(RNAi)* and control worms differed significantly (p < 0.001, non-linear regression analysis). Scale bar, 50 μm on left panels, 10 μm on right panels. All counts were performed in 5–10 animals.

The following figure supplements are available for figure 6:

**Figure supplement 1**. *mex3-1* RNAi impairs brain differentiation.

**Figure supplement 2**. Differentiation in *p53(RNAi)* animals.

cells, we quantified the proportion of *piwi-1⁻*PIWI-1⁺ cells, which are thought to represent immediate stem cell progeny that have permanently exited the cell cycle. We found that *mex3-1* RNAi resulted in a significant increase in the number of *piwi-1⁺*PIWI-1⁺ cells and simultaneous decrease in *piwi-1⁻*PIWI-1⁺ cells compared to controls (*Figure 6F*), supporting a general reduction in the ability of stem cells to progress to a postmitotic state. From these data demonstrating abrogated production of lineage-restricted stem cell progeny, impaired contribution to the turnover of multiple tissue types, and concomitant increases in all known stem cell subclasses, we propose that *mex3-1* is a critical regulator for all differentiating progeny, mediating the adoption of a non-stem cell fate.

### RNAseq of *mex3-1(RNAi)* animals identifies novel progenitor transcripts

RNAseq of *mex3-1(RNAi)* whole animals 12 days after RNAi was performed to provide a broad and comprehensive overview of gene expression changes. Given that *mex3-1* RNAi specifically eliminates postmitotic progeny fates, we reasoned that this approach may offer a more selective method than X2-FACS enrichment in order to identify additional progenitor-specific transcripts both within and outside the epithelial lineage. In agreement with our data demonstrating hyper-proliferation and expansion of the entire stem cell compartment after *mex3-1* knockdown, 13/59 known stem cell-specific transcripts were significantly upregulated upon knockdown of *mex3-1* (p < 0.01, *Figure 7—figure supplement 1A*, *Supplementary file 4*). These included cell cycle genes, two of which were further confirmed by WISH (*PCNA* and *H2B*; *Figure 7—figure supplement 1B*). Importantly, we also observed an approximately 1.5-fold increase in *soxP-1*, *zfp-1*, and *piwi-1* transcripts in *mex3-1(RNAi)* animals (*Figure 7—figure supplement 1A*; *Supplementary file 4*), concordant with the increase in stem cell subclasses quantified by cell counting (*Figure 5C,D*).

As anticipated from the effects of *mex3-1(RNAi)* on the loss of all epithelial progenitor markers by WISH analyses, all but two of our new early and late progeny markers were severely down-regulated in the RNAseq data set for *mex3-1(RNAi)* animals compared to controls (*Figure 7A*; *Supplementary file 4*). The two transcripts that did not appreciably change (*sox9-1* and *RPC-2*) have very high X1 expression in addition to labeling epithelial progenitors, which was not expected to change in *mex3-1* RNAi. Together, the RNAseq data corroborate the in vivo cell type analyses, where *mex3-1* was required to restrict the stem cell compartment and promote differentiation of progenitor fates. We next tested whether transcripts down-regulated following *mex3-1* RNAi mark novel neoblast progeny.

We chose 21 down-regulated and uncharacterized transcripts for validation by WISH. By RNAseq, all were irradiation-sensitive but not X2-enriched (*Figure 7A*, *Supplementary file 4*) and thus could be classified as WT^high^X^low^. We found that 20/21 transcripts produced a *prog*-like pattern in control worms and exhibited severely reduced expression in *mex3-1(RNAi)* worms, as anticipated from RNAseq (*Figure 7B*, *Figure 7—figure supplement 1C*). The remaining transcript, *Sme-dASXL_059179*, was highly expressed near the proximal end of the pharynx, throughout the pharynx proper, and was similarly dependent on *mex3-1* for its expression (*Figure 7B*). Three transcripts were confirmed to be irradiation-sensitive by WISH (*Figure 7—figure supplement 2A*), and two of these were verified to be additional late progeny markers based on their high degree of co-localization with *AGAT-1* expression (*Figure 7—figure supplement 2B*). We predicted that *SmedASXL_059179⁺* cells may represent a pharyngeal progenitor cell type, and indeed, we observed PIWI-1⁺ *SmedASXL_059179⁺* cells near the proximal base of the pharynx (*Figure 7C*). However, no regulatory roles were uncovered for this gene, as RNAi against *SmedASXL_059179* did not result in apparent perturbations to pharynx function during homeostasis or regeneration (*Figure 7—figure supplement 2C,D*). Overall, these results further support a role for *mex3-1* as a critical regulator of differentiation toward multiple lineages and also demonstrate the utility of transcriptional analysis of *mex3-1(RNAi)* in identifying additional markers of tissue-specific progenitor populations, as an alternative to the criteria of irradiation-sensitivity and FACS localization.

## Discussion

### The X2 population has high heterogeneity and a strong epithelial progenitor gene signature

At the outset of this study, most efforts had been put forth to understand stem cell (X1) transcriptomes, heterogeneity, and genetic regulators. Relatively little was known about the X2 FACS cell fraction, with the exception that it expresses a disproportionate number of transcription factors

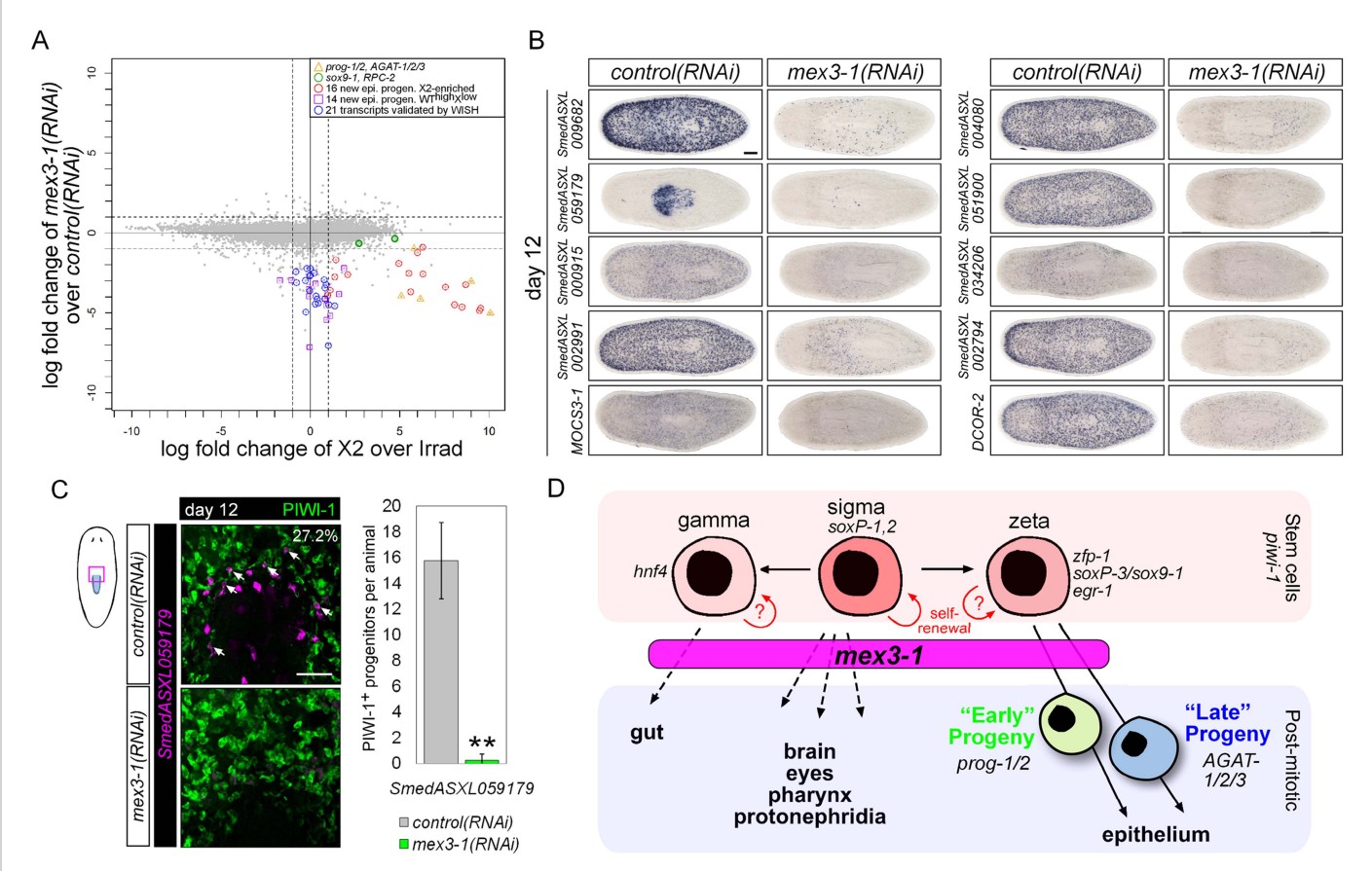

**Figure 7**. Transcriptional analysis after *mex3-1* RNAi identifies novel progenitor transcripts. (**A**) RNAseq was performed on *mex3-1(RNAi)* animals 12 days after RNAi. Each gray dot represents one transcript. Established and newly identified progeny markers are indicated. Top *mex3-1* down-regulated transcripts cloned out for validation by WISH are indicated as well. (**B**) The blue-circled transcripts from (**A**) all require *mex3-1* for expression. Scale bar, 200 μm. (**C**) Assessment of the *mex3-1(RNAi)*-down-regulated transcript *Smed_ASXL059179* as a marker for a novel pharynx progenitor cell type using FISH and PIWI-1 immunolabeling. Confocal projections in control and *mex3-1* RNAi animals are shown. Error bars represent standard deviation. Scale bar, 50 μm. **p < 0.01 (Student's *t*-test). (**D**) Model of lineage specification in planarian stem cells. *mex3-1* is a key determinant in balancing stem cell self-renewal and differentiation, acting as a promoter of postmitotic cell fates and commitment toward multiple lineages.

The following figure supplements are available for figure 7:

**Figure supplement 1**. RNAseq analysis of *mex3-1(RNAi)* animals.

**Figure supplement 2**. Characterization of down-regulated genes in *mex3-1(RNAi)* animals.

and shares most of its gene signature with the X1 fraction, and the few known stem cell progeny markers are most highly expressed in this fraction (*Labbe et al., 2012*). We reasoned here that investigating more genes specific to the X2 fraction could yield many markers and regulators for other tissue-specific progeny types. We found that transcripts enriched in the progeny-associated X2 FACS gate were expressed in a variety of cell types and not necessarily in stem cell progeny. Although transcripts with unique tissue-specific and non-stem cell expression patterns were uncovered and appeared to be potential candidates for novel progenitor types, it is now clear that the epithelial lineage has the strongest bias in terms of sequencing numbers. In total, we describe 32 new markers for early and late epithelial progenitors. Combined with the revelation that zeta-neoblasts are the largest known lineage-restricted stem cell subclass in planarians, we conclude that epithelial

progenitors are the predominant output of stem cells, most likely reflecting highest turnover needs as well as molecular complexity of this organ.

## *mex3-1* as a candidate mediator of asymmetric cell fate in planarians

Through RNAi screening of the 120 candidate progeny transcripts, we identified *mex3-1* as a critical factor in cell fate decision-making. *mex3-1* was required not only for all epithelial progenitor fates but also all other tested lineage-restricted stem cell progeny, and to restrict the expansion of the stem cell compartment. MEX3 is an RNA-binding translational repressor discovered in *C. elegans* with embryonic and post-embryonic developmental roles (*Draper et al., 1996*). MEX3 is a fundamental regulator of asymmetry, mediating one of the earliest steps of cell fate determination in *C. elegans* by restricting expression of the cell fate determinant *pal-1* transcription factor to the posterior blastomere of the early embryo (*Draper et al., 1996*; *Huang et al., 2002*). In adult *C. elegans*, MEX3 participates in the maintenance of germline stem cells (GSCs) by promoting proliferation, and interestingly, is not required for GSCs to differentiate and undergo meiosis (*Ariz et al., 2009*). Four *mex3* homologs (*MexA-D*) are present in vertebrates, which remain poorly characterized (*Buchet-Poyau et al., 2007*). Recent work on MEX3A in murine intestinal cell culture showed that the master intestinal differentiation factor *Cdx2*, one of the vertebrate homologs of *pal-1*, is a target of MEX3A translational inhibition, and furthermore, MEX3A overexpression increased levels of intestinal stem cell markers such as *Bmi1* and *Lgr5* (*Pereira et al., 2013*). Though the role of MEX3A in intestinal stem cell fate in vivo remains to be fully explored, these findings combined with our results in planarians suggest a conserved ancestral function for *mex3* genes in cell fate choice.

The precise mechanisms by which *mex3-1* exerts its effects in planarians are unresolved. Although both *zfp-1* and *mex3-1* are required for specification of epithelial-fated progeny, unlike *zfp-1(RNAi)* worms, *mex3-1(RNAi)* worms completely lose the ability to form any regenerative blastema. *mex3-1* was further needed to prevent the expansion of the stem cell compartment, including sigma-, gamma-, and zeta-class neoblasts, which suggests that *mex3-1* acts to mediate stem cell lineage asymmetry in the general stem cell pool and regulate all postmitotic lineages. As both *C. elegans* and vertebrate MEX3 proteins have been shown to be translational repressors, it is likely that this molecular function has been conserved in planarians. Thus, we propose a model of stem cell lineage progression with *mex3-1* acting as a repressor of stem cell identity and self-renewal genes in postmitotic progenitors to promote differentiation (*Figure 7D*).

During early *C. elegans* embryogenesis, the asymmetric distribution of both *mex3* mRNA and protein underlies the asymmetric expression of *pal-1* (*Draper et al., 1996*). While *Smed-mex3-1* mRNA shows no such asymmetry at the current resolution of our tools in planarians, it is possible that asymmetric distribution or activation of MEX3-1 leads to the execution of its function in progeny only. An example of such a mechanism is Prospero in *Drosophila* neuroblasts, where it is transcribed and translated in stem cells and daughter cells, but is segregated to and functions in progeny in a classic example of intrinsic asymmetric cell division (*Doe et al., 1991*). Alternatively, MEX3-1 may have multiple cell type-specific roles and cell type-specific mRNA targets. No planarian homolog to the conserved MEX3 target *Cdx2*/*pal-1* could be found in existing transcriptomes or genomic sequence (*Sánchez Alvarado et al., 2003*; *Robb et al., 2008*; *Abril et al., 2010*; *Sandmann et al., 2011*; *Labbe et al., 2012*; *Onal et al., 2012*; *Resch et al., 2012*; *Solana et al., 2012*; *Fernandes et al., 2014*). Therefore, elucidating the RNA targets of MEX3-1 will provide an opportunity to uncover novel RNA targets in other organisms and ASC systems.

## Finding progenitors of other cell types

Now that we have identified additional epithelial progenitor markers and progeny regulators, the question then becomes, what it is the best method to discover non-epithelial progenitors of other tissue types in planarians? With the evidence described in this study implicating *mex3-1* as a regulator of multiple neoblast progeny for other tissues, transcriptional analysis of *mex3-1(RNAi)* animals may be a promising avenue to discover rare and novel progeny markers. For instance, comparing the wild-type X2 fraction with X2 cells isolated from *mex3-1(RNAi)* worms could provide insight into other cell types lost in this FACS gate due to specific cessation of differentiation, without confounding effects of irradiation or general ablation of the stem cell pool.

In addition to the assumption that differentiating progeny cells exist in the X2 fraction, it is clear that there are irradiation-sensitive transcripts and cells outside of the X1 and X2 FACS gates. At this juncture, isolating these populations from irradiation-insensitive cells is not achievable as there are no clear boundaries on where these cells would lie in FACS plots. Numerous transcripts, down-regulated after *mex3-1* RNAi, were observed to belong to this WT$^{high}$X$^{low}$ category with little expression in the X1 or X2 fractions. The identification of one of these transcripts as a pharyngeal marker with expression in PIWI-1$^+$ cells demonstrates that lineage-restricted progeny are present outside the X1 and X2 gates and merits the investment of future efforts toward screening further WT$^{high}$X$^{low}$ genes that are regulated by *mex3-1* in order to uncover novel progenitors. This knowledge is necessary in order to achieve a comprehensive understanding of the asymmetries specific to ASC lineage organization. Due to the conserved role of MEX3 in regulating cell fate determination across multiple phyla, understanding how *mex3-1* achieves lineage asymmetry in planarians will contribute to informing ASC biology and regeneration in other organisms.

## Materials and methods

### Deep sequencing

We previously performed RNAseq of the planarian X1, X2, and irradiation-insensitive compartments where the X2 cell fraction was sequenced to a depth of 206 million reads in two biological replicates (*Labbe et al., 2012*). Here, we performed an additional replicate by using flow cytometry to obtain cell populations as previously described (*Hayashi et al., 2006*; *Pearson and Sánchez Alvarado, 2010*). Approximately, 1 million X2 cells from 100 animals were isolated on a Becton–Dickinson FACSaria over multiple sorts. Total RNA was purified and poly-A-selected cDNA libraries were prepped using the TruSeq kits from Illumina. This new X2 sample was multiplexed together with a new X1 and Irradiated control and each was sequenced to a depth of >63 million single-end 50 base pair reads on an Illumina HiSeq2500 with v4 chemistry. Raw sequence data were uploaded to NCBI GEO under accession number GSE68581. Each sample was aligned to the transcriptome under NCBI BioProject PRJNA215411 using Bowtie2 with no sequence trimming. Mapped reads per million reads (CPM) of each transcript was calculated. Note that kilobase-length of each transcript was not taken into account because in any expression ratio, the length scaling factor cancels out and no inter-transcript comparisons were performed. To ensure a well-defined statistic in the calculation of fold-change, pseudocounts of +1 were added to every numerator and denominator as a way to not bias differentially expressed genes toward lowly expressed transcripts (*Klattenhoff et al., 2013*). The transcripts listed in *Supplementary files 1–4* can be found in the same transcriptome database. The heatmap in *Figure 1C* was created using the Partek Genomics Suite of software (www.partek.com) with the unsupervised hierarchical clustering algorithm: Pearson's Absolute Value Dissimilarity. Intact unirradiated control (WT) transcript levels were averaged from 12 replicates of unfed and *control(RNAi)* experiments and time points using over 700 million sequencing reads (*Labbe et al., 2012*; *Solana et al., 2012*; *Currie and Pearson, 2013*; *Zhu and Pearson, 2013*; *Lin and Pearson, 2014*).

### Analysis of deep sequencing data

In order to validate the consistency of our previous and new deep sequencing replicates, Pearson correlations were performed with our own data as well as all previously published RNAseq relevant to the current study using CPM for each transcript with CPM <1000 (*Figure 1—figure supplement 1*) (*Labbe et al., 2012*; *Onal et al., 2012*; *Resch et al., 2012*; *Solana et al., 2012*). The program DESeq was used to determine significantly enriched transcripts in the X2 (FDR < 0.01) or X1 (FDR < 0.001) cell fractions vs irradiated whole animals using the three biological replicates for each tissue type. MA plots, Pearson correlations, and log fold change plots were made using *R*.

### Phylogenetics and cloning

Transcripts identified by differential expression were cloned using forward and reverse primers into a double-stranded RNA expression vector as previously described (*Rink et al., 2009*). Riboprobes were made from PCR templates from the same vector (pT4P) (*Pearson et al., 2009*). The three MEX3

homologs in *S. mediterranea* were identified with tBLASTn searches of the planarian genome/transcriptomes using MEX3 protein sequences from *C. elegans* and mouse. Candidate planarian MEX3 homologs were validated by reciprocal BLASTx against the nr database (NCBI). The predicted proteins of the planarian MEX3 homologs were aligned using the program T-coffee along with MEX3 homologs from other species (*Notredame et al., 2000*). The program Geneious (www.geneious.com) was used to run a Bayesian phylogeny using the MrBayes plugin with the following settings: a WAG substitution model, 25% burnin, subsample frequency of 1000, 1 million replicates, and four heated chains (*Ronquist and Huelsenbeck, 2003*). The transcripts for *mex3-1* and all new progeny markers from this manuscript are listed in *Supplementary file 2*.

## Animal husbandry and RNAi

Asexual *S. mediterranea* CIW4 strain was reared as previously described (*Sánchez Alvarado et al., 2002*). RNAi experiments were performed using previously described expression constructs and HT115 bacteria (*Newmark et al., 2003*). Briefly, bacteria were grown to an O.D. 600 of 0.8 and induced with 1 mM isopropyl β-D-1-thiogalactopyranoside (IPTG) for 2 hr. Bacteria were pelleted and mixed with liver paste at a ratio of 500 µl of liver per 100 ml of original culture volume. Bacterial pellets were thoroughly mixed into the liver paste and frozen as aliquots. The negative control RNAi was the *unc22* sequence from *C. elegans* as previously described (*Reddien et al., 2005a*). For the screening of all genes in this study, RNAi food was fed to 7-day starved experimental worms every third day for five feedings. Subsequent functional analyses for *mex3-1* were performed with one feed unless noted otherwise. Time points in figures denote the number of feeds for each gene as well as the number of days after the last feed. For example, 1fd12 corresponds to one RNAi feeding and 12 days after that feeding. Amputations were performed 6 days after the final feeding unless noted otherwise. All animals used for immunostaining were 3–4 mm in length and size-matched between experimental and control worms.

## Immunolabeling, TUNEL, EdU, BrdU, irradiation, and WISH

WISH, dFISH, and immunostaining were performed as previously described (*Pearson et al., 2009*; *Lauter et al., 2011*; *Cowles et al., 2013*; *Currie and Pearson, 2013*). Colorimetric WISH and fluorescent phospho-histone H3 (H3P) stains were imaged on a Leica M165 fluorescent dissecting microscope. The rabbit monoclonal antibody to H3ser10p from Millipore (04–817) was used for all cell division assays (*Newmark and Sánchez Alvarado, 2000*). TUNEL was performed as previously described (*Pellettieri and Sánchez Alvarado, 2007*). Mouse anti-PIWI-1 (gift of Dr Jochen Rink [*Wagner et al., 2011*]) was used at 1:1000. H3ser10p and TUNEL were quantified using freely available ImageJ software (http://rsb.info.nih.gov/ij/). Significance was determined by a 2-tailed Student's *t*-test unless otherwise noted. All experiments were, at minimum, performed in triplicate with at least 10 worms per stain and per time point (i.e., n > 30). For irradiation experiments, planarians were exposed to 16.5 or 60 Gy of γ-irradiation from a $^{137}$Cs source. F-*ara*-EdU was fed to worms in liver paste at a concentration of 0.05 mg/ml for a 7-day chase (fed at 1fd6) or 0.5 mg/ml for a 24-hr chase (fed at 1fd6 and 1fd9) and stained as previously described following the normal fixation for ISH (*Pearson et al., 2009*; *Neef and Luedtke, 2011*). BrdU was fed (at 1fd6) in liver paste at a concentration of 10 mg/ml and stained as previously described (*van Wolfswinkel et al., 2014*). Confocal images were acquired on a Leica DMIRE2 inverted fluorescence microscope with a Hamamatsu Back-Thinned EM-CCD camera and spinning disc confocal scan head, and stitched together for whole-animal images. Images were post-processed in Adobe Photoshop and figures assembled in Macromedia Freehand.

## Acknowledgements

We thank Drs George Eisenhoffer, Jason Pellettieri, Ricardo Zayas, and Ian Scott for helpful comments on the manuscript. We thank Dr Marc Remke for assistance with the heatmap. BJP and SEH were funded by Ontario Institute for Cancer Research Grant #IA-026. SJZ was funded by a RestraComp student fellowship at The Hospital for Sick Children. KWC was funded by NSERC grant #402264-2011.

## Additional information

### Funding

| Funder | Grant reference | Author |
|---|---|---|
| Ontario Institute for Cancer Research | IA-026 | Stephanie E Hallows, Bret J Pearson |
| Natural Sciences and Engineering Research Council of Canada | 402264-2011 | Ko W Currie |
| The Hospital for Sick Children | Restracomp | Shu Jun Zhu |

The funders had no role in study design, data collection and interpretation, or the decision to submit the work for publication.

### Author contributions

SJZ, Conception and design, Acquisition of data, Analysis and interpretation of data, Drafting or revising the article; SEH, KWC, Conception and design, Acquisition of data, Analysis and interpretation of data; CJX, Conception and design, Analysis and interpretation of data; BJP, Conception and design, Drafting or revising the article

## Additional files

### Supplementary files

• Supplementary file 1. RNA-deep sequencing (RNAseq) from this study and previously published studies and analysis. Raw reads from RNAseq of fluorescence-activated cell sorting (FACS)-isolated X1 and X2 populations, lethally irradiated (60 Gray) whole worms (Irrad), and intact control worms (WTcontrols). *Tab 2*, transcripts identified using DESq as having enriched expression in the X1 population compared to irradiated animals, with adjusted *p*-value < 0.001. *Tab 4*, transcripts identified using DESq as having enriched expression in the X2 population compared to irradiated animals, with adjusted *p*-value < 0.01.

• Supplementary file 2. Established stem cell and progeny genes, and genes characterized in this study. *Tab 1–2*, listing of 59 verified neoblast-specific genes (*Known stem cell genes*) and five established postmitotic progeny genes previously shown to be enriched in the X2 FACS gate (*Known X2 epithelial progenitor genes*) used in *Figure 1*, *Figure 7*, and *Figure 7—figure supplement 1*. *Tab 3*, annotation of the top 100 X2-enriched genes chosen for expression analysis and RNA interference (RNAi) screening in this study. *Tab 4*, a total of 66 transcripts were identified as WT$^{high}$X$^{low}$, which are irradiation-sensitive transcripts with low expression in both the X1 and X2 irradiation-sensitive FACS gates. *Tab 5*, epithelial progenitor markers identified in this study.

• Supplementary file 3. Cell counts for co-localization of identified progeny markers with established epithelial progenitor genes. Total cell counts for established lineage markers *piwi-1*, *prog-1*, *prog-2*, and *AGAT-1*, and new progeny markers identified in this study are shown. All counts were performed in 3–4 animals at multiple body regions (head, pre-pharyngeal, tail).

• Supplementary file 4. Transcriptional analysis of *mex3-1(RNAi)* animals. Raw RNAseq reads of control and *mex3-1(RNAi)* animals are shown. Best homology to fly and mouse proteins are shown for the top 100 down-regulated genes. The top 21 uncharacterized down-regulated genes chosen for validation by WISH are indicated.

## Major datasets

The following dataset was generated:

| Author(s) | Year | Dataset title | Dataset ID and/or URL | Database, license, and accessibility information |
|---|---|---|---|---|
| Zhu SJ, Hallows SE, Currie KW, Xu C, Pearson BJ | 2015 | A mex3 homolog is required for differentiation during planarian stem cell lineage development | http://www.ncbi.nlm.nih.gov/geo/query/acc.cgi?acc=GSE68581 | Publicly available at the NCBI Gene Expression Omnibus (Accession no: GSE68581). |

Standard used to collect data: NCBI GEO

The following previously published datasets were used:

| Author(s) | Year | Dataset title | Dataset ID and/or URL | Database, license, and accessibility information |
|---|---|---|---|---|
| Pearson BJ | 2012 | Planarian stem cell transcriptome | http://www.ncbi.nlm.nih.gov/geo/query/acc.cgi?acc=GSE37910 | Publicly available at the NCBI Gene Expression Omnibus (Accession no: GSE37910). |
| Solana et al, | 2012 | RNAseq in H2B RNAi | http://sra.dnanexus.com/studies/ERP001079/runs | European Sequence archive. |
| Alvarado S | 2015 | Schmidtea mediterranea CIW4 Transcriptome | http://www.ncbi.nlm.nih.gov/bioproject/?term=PRJNA215411 | Publicly available at the NCBI Gene Expression Omnibus (Accession no: PRJNA215411). |

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
