## [Decision Letter]

Thank you for sending your work entitled “A *mex3* homolog drives differentiation during planarian stem cell lineage development” for consideration at *eLife*. Your article has been favorably evaluated by Fiona Watt (Senior editor), Marianne Bronner (Reviewing editor), and three reviewers.

The Reviewing editor and the reviewers discussed their comments before we reached this decision, and the Reviewing editor has assembled the following comments to help you prepare a revised submission.

The authors transcriptionally profiled a population of irradiation-sensitive G1/G0 cells in planarians (from the “X2” FACS gate) that is a mixture of cycling and immediately post-mitotic cells. Analysis of genes enriched in this population identified 32 new markers of postmitotic neoblast progeny differentiating into epidermal fates. Additionally, through RNAi screening of these genes, *mex3-1* was identified as essential for differentiation into multiple planarian cell lineages (eye, neuronal lineages, and epithelium). *Mex3-1* inhibition decreased numbers of post-mitotic progenitors cells and concomitantly increased numbers of proliferating neoblasts, including three lineages specified within that population, causing failure of regeneration and tissue homeostasis. RNA-seq from *mex3-1(RNAi)* animals confirmed that *mex3-1* is a master regulator of genes expressed in post-mitotic neoblast progeny and also identified novel markers of differentiating cells in planarians. MEX3 proteins can repress translation to influence differentiation and asymmetric cell division. Still few molecules have been identified that control differentiation and cell cycle exit in planarians, and the identification of *mex3-1* in particular suggests the essentiality of asymmetric cell division in forming progenitor populations.

Although symmetric and asymmetric divisions have been predicted in this organism, this study provides the first molecular link to those processes in planarians. However, there are significant points that need to be addressed.

1) The authors need to clarify the role of *mex3* in cell cycle exit by determining the ratio of *piwi*(mRNA)+*piwi*(protein)+ vs. *piwi*(mRNA)-*piwi*(protein)+ cells after *mex3* RNAi.

2) They should address how *mex3(RNAi)* is distinct from RNAi of other factors required for neoblast differentiation (i.e. *p53*). In particular, does loss of *mex3* result in eventual loss of the neoblasts like *p53*? In Figure 6 there is a ∼3 fold drop in pH3+ cells between days 12 and 15. Is this due to depletion of the neoblast pool, similar to what happens with *p53*? What about at day 18? The authors should also assess *p53* requirement in gut/neural progenitors to determine if *mex3* is broader in its role.

3) The authors need to provide further quantification of the neoblast results. The model that *mex3* is promoting the differentiation/renewal balance is influenced from their data on neoblast numbers, but at present, the data are not sufficient to strongly make the case. The EdU quantification indicated that there are more S-phase cells in *mex-3* RNAi animals, but this is a technically limited experiment because of patchy labeling. Where the counting was done in even an individual animal would yield very different results. Furthermore, the FACs quantification of cells greater than 2C (X1s) shows no clear difference or perhaps less in *mex3* RNAi (Figure 4—figure supplement 1), and no increase in expansion of neoblasts, compared to the control, in the sublethal irradiation experiment is apparent. Figure 5 provides the strongest support, but no numbers are given, and it would be stronger with quantification from many animals and at multiple AP planes.

4) Along the same lines, using the PIWI antibody (Figures 5 and 6) labels neoblasts and their descendants. Therefore, technically, stem cell subclasses are not shown. PIWI+ *hnf4*+ cells increase following *mex3* RNAi, whereas *ovo+* or *coe+* PIWI+ cells decrease. How do these opposing results fit with the model? The authors should attempt to resolve quantitative impacts of *mex3* RNAi on neoblast subclasses versus their non-dividing descendants using marker/*piwi*/PIWI labelings.

5) Figures/data:

Figure 1—figure supplement 1: It is unclear whether the correlations were done on raw values or log-transformed values; if not log-transformed analyses, it should be plotted and analyzed this way to prevent overestimation of the correlation. Range/units on X and Y axes should be present. Graphs are overplotted, and might be better represented with 2D density plots. Red line is unclear; representing “a Pearson correlation coefficient of 1” is unclear. The correlation P values are unnecessary. RPM should be changed to CPM.

Figure 1: Y axis states “over Xins” when the comparison is to whole worms and Xins has been used to refer to a FACs gate in the literature.

In the first paragraph of the subsection headed “RNAseq analysis of the progeny-associated X2 FACS cell fraction”: “stem cell markers can be highly expressed in the X2 fraction (Figure 1)” – this is not shown in Figure 1.

Figure 1: The colors are saturated, and it seems the range shown does not capture the range of the data. If they are setting a maximum/minimum for the graph that is below the range of the data this should be stated in the legend, or the shown range changed.

Figure 1: The “Lowess” curve (Loess fit?) is not needed, and the authors are encouraged to remove it.

Figure 1: “new progeny markers” is an awkward Figure label for these genes.

Figure 1 legend: the description is poor – it is a heatmap, Z score range, etc.

Figure 1—figure supplement 2 is hard to read; trimming the end/uninformative part(s) would help focus on the region with similarity. Shown is not a “MUSCLE alignment” but a multiple sequence alignment done with MUSCLE. There is more than one shade of blue, which either represent similar or identical aa. The three rows of information below the alignment should be better described. Why not show all 24 in the alignment?

Figure 2, genes are divided into early and late, but a given late marker in principle could also have been present in early cells. This might be worth noting in the text.

In the first paragraph of the subsection headed “New progeny markers label either early or late progeny of the epithelial lineage” the authors find that 5.4% of *prog-1* cells expressed low levels of AGAT-1, and Eisenhoffer 2008 reported 44%. They refer to these populations as “two mainly non-overlapping”, which needs some clarification.

Figure 3.

Part A might be better as time following initiation of RNAi.

Part E: what happens at longer time points post RNAi? How much expression is not irradiation sensitive?

*mex3* orthology: The tree lacks an outgroup(s) with a similar RNA-binding domain to that present in this protein family. Therefore, without outgroups, any non-*mex3* family protein would give the same result as planarian MEX3 as presented in the tree in Figure 3—figure supplement 1.

Figure 4.

C: there is a clear TUNEL effect, without clear explanation. It might be worth simply noting in the text that the explanation for this late rise in TUNEL+ cell number is unclear.

C/D: Where counts were performed should be stated.

The X2s by FACs don't seem to be decreased following mex RNAi (Figure 4—figure supplement 1), and yet there is a strong impact on *prog*-like+ cells by in situ. Is there an explanation? For example, is the fraction of X2s that are neoblasts (*piwi+*) increasing?

Figure 5.

B: This is a nice result clearly showing neoblast increases can occur in *mex3* RNAi animals. The images are dark and hard to see in the present figure file.

D: Using the PIWI antibody labels neoblasts and their descendants. Therefore, technically stem cell subclasses are not shown. I recommend having a simple cell count plot instead of showing ratios. The high variance of the data in the *hnf4* and *zfp1* counts, suggest the the *p*-value is a bit surprising, although without seeing the counts, that is hard to evaluate.

Figure 6.

A: There wasn't really a “chase”, it was a pulse + 7 days. The wording could be modified.

C: See comments on Figure 5. The PIWI ab here is less of a concern, if the wording is clear that these are neoblasts and/or descendants involved in tissue turnover. The ratio comment applies here as well. Number of animals counted should be noted.

Is *mex-3* expressed in the any/all of these three sets of cells?

Figure 7.

B: Are these indeed a similar class of cells studied in prior parts of the paper? (Several double-FISH tests would quickly confirm this). How do these look in the WT-high X-low data?

C: numbers, rather than ratios would give reader a better understanding of how many cells were present/ counted. What fraction of these are PIWI+?

What is the expression of the candidate pharynx progenitor marker in irradiated worms?

[Editors' note: further revisions were requested prior to acceptance, as described below.]

Thank you for resubmitting your work entitled “A *mex3* homolog is required for differentiation during planarian stem cell lineage development” for further consideration at *eLife*. Your revised article has been favorably evaluated by Fiona Watt (Senior editor), a Reviewing editor, and three reviewers. The manuscript has been improved but there are some remaining issues that need to be addressed before acceptance, as outlined below:

1) Reply to point #2: “We have included data demonstrating that *mex3-1* is also expressed in *chat*^*+*^ neural progenitors (Figure 6—figure supplement 1)” – this was done with PIWI1 antibody and therefore these cells are not necessarily progenitors; they could be newly differentiated cells. “*p53* knockdown does not significantly reduce *ovo*^*+*^ and *chat*^*+*^ progenitors” – again, it was done with a PIWI1 antibody, and therefore those are not necessarily progenitors.

The wording should be addressed accordingly (or *smedwi* mRNA could be used).

Related to this, in the reply to major point 4: Would the authors’ argument about *hnf4* predict that *smedwi* (mRNA)+ *coe*+ cell would go up in frequency? This could in principle be tested to further support the proposed model.

2) Figure 1—figure supplement 1. Again, in this reviewer's view plots should be log transformed and correlation done on log-transformed values.

3) Figure 1. The response did not satisfactorily address the review comment. The legend states the z-score range shown, but if the data is not showing the actual z-score values, why not state what the actual range of the z-scores was in the data in the legend such that it is clear that some values were much lower/higher than the cutoffs on the range shown?

4) MEX-3 orthology: The authors clarified their goal for the MEX3 phylogeny, which makes this tree of limited value; however, the answer leaves the evidence for MEX-3 orthology poorly described in the manuscript. Perhaps the authors can more explicitly describe in the methods the evidence that this is a MEX-3 ortholog (e.g., phylogenetics, unique domain architecture specific to this family, or lack of any other proteins with similarity near that present with the MEX-3 family in reciprocal genome searches).

[Editors' note: final revisions were requested prior to acceptance, as described below.]

Thank you for resubmitting your work entitled “A *mex3* homolog is required for differentiation during planarian stem cell lineage development” for further consideration at *eLife*. Your revised article has been evaluated by Fiona Watt (Senior editor), a Reviewing editor, and one of the original reviewers. The manuscript has been improved but there are still some differences in interpretation that we would ask you to address.

Reviewer #2:

Below is a discussion on comments regarding SMEDWI^+^ cells and the term “progenitor”.

1) Original Review: “We have included data demonstrating that *mex3-1* is also expressed in *chat*^*+*^ neural progenitors (Figure 6—figure supplement 1)”. This was done with PIWI1 antibody and therefore these cells are not necessarily progenitors; they could be newly differentiated cells.

Author Comment: “We believe this is confusion in terminology. By ‘progenitors’ we do not mean an equivalent of dividing progenitors or transit amplifying cells from other systems, but instead imply a committed cell that has not yet fully differentiated and physically integrated into the terminal tissue…”

Reviewer Reply: Dividing or being transit amplifying is not the point of confusion. It is simple: a SMEDWI^+^ cell could be differentiated. SMEDWI^+^ cells can be fully integrated into tissues and express differentiated markers. For instance, [83]: “SMEDWI-1^+^ descendant cells expressed a choline acetyl-transferase ortholog, Smed-chat (Figure supplement 6); *chat* expression is widely conserved in cholinergic neurons (34). SMEDWI-1+; *chat*^*+*^ cells were enriched in brain regions and had neuronal morphology, and *chat*^*+*^ cells co-expressed additional neuronal markers (Figure supplement 6), indicating that SMEDWI-1^+^; *chat*^*+*^ cells are differentiating neurons”.

Similarly, [73]: SMEDWI^+^ cells can also be seen to co-express protonephridial differentiated cell markers (carbonic anhydrase and cubulin).

2) Author Comment: “To try to minimize this confusion, we named most of our newly discovered transcripts in the ‘postmitotic progeny’ series. In this current revision, we have also tried to minimize the use of the word progenitor. So to directly address this comment: ‘progenitors’, ‘stem cell progeny’, and ‘newly differentiating cells’ are equivalent”.

Reviewer Reply: “newly differentiated cells” is what was written, but “newly differentiating” and “newly differentiated” are hard to really to distinguish regardless. The point is that calling something a neural progenitor, when it might be a cell with axons and differentiating or differentiated seems inaccurate.

3) Author Comment: “Because PIWI-1 labels stem cells and immediate post-mitotic descendants, we consider ‘progenitors’ to be cells that are PIWI-1+[tissue-specific marker]+, to mark cells in the early stages of commitment”.

Reviewer Reply: In this reviewer’s view, this cannot be concluded (see above). According to available data, it is not possible to exclude, without additional data for a given tissue/cell type, the possible interpretation that any given SMEDWI^+^ cell is a differentiated cell.

4) Author Comment: “As no permanent lineage tracing can yet be done in our system, it is impossible to know the terminal fates of individual cells and whether these *piwi-1*^*+*^ progenitors self-renew. We have attempted to clarify this in the text, and this rebuttal document will also help readers”.

Reviewer Reply: The request was simply to be clear with wording in the instances when PIWI antibody is used only. There is strong evidence that these PIWI+ cells are neoblast descendants, either undifferentiated or newly differentiated. It can be useful as a tool used in this way; the authors should just be clear with wording, describing these cells as neoblast descendants rather than progenitors. Or, one could choose to be more explicit than “neoblast descendant” and state: neoblast, undifferentiated neoblast descendants, or newly differentiated/ing neoblast descendants. Or, if “progenitor” was really desired, something along the lines of “neoblast descendants, including undifferentiated progenitors and possibly newly differentiated cells”.

5) Original Review: “*p53* knockdown does not significantly reduce *ovo*^*+*^ and *chat*^*+*^ progenitors” – again, it was done with a PIWI1 antibody, and therefore those are not necessarily progenitors. The wording should be addressed accordingly (or *smedwi* mRNA could be used).

Author Comment: “We have clarified in the text our use of the term ‘progenitor’. We use the same progenitor term used in Lapan and Reddien (Plos Genetics 2011) and Lapan and Reddien (Cell Reports 2012) describing the *sp6-9*^*+*^ and *ovo*^*+*^ cells as eye progenitors, respectively. A similar terminology was used in the *eLife* manuscript by [3] regarding *FoxA*^*+*^ progenitors. Text changes can be found in the third paragraph of the Introduction and in the subsection headed ‘*mex3-1* is required for turnover of multiple tissues’.”

Reviewer Reply: In the case of the eye, cells outside of the eye (an epithelial optic cup and a neural ganglion) could be visualized and described as progenitors. A SMEDWI^+^ neuron in the eye would not considered a progenitor, but an eye cell – a similar such cell would be harder to identify in a less discrete tissue/cell population.

---

## [Author Response]

*1) The authors need to clarify the role of* mex3 *in cell cycle exit by determining the ratio of* piwi*(mRNA)+*piwi*(protein)+ vs.* piwi*(mRNA)-*piwi*(protein)+ cells after* mex3 *RNAi*.

This is a good experiment and we now provide these data and show that the ratio of *piwi-1*^+^PIWI-1^+^ cells to *piwi-1*^-^PIWI-1^+^ cells increases significantly after *mex3-1* RNAi (Figure 6). Counts were performed in multiple regions of animals (head, neck, and tail). The percentage of PIWI-1^+^ cells negative for *piwi-1* mRNA is 22.7% in controls, and 6.4% in *mex3-1(RNAi)* animals, further supporting the role of *mex3-1* in promoting the global adoption of a post-mitotic fate. It should be noted that this suggested experiment has 2 rather large assumptions. First, that *piwi-1* RNA is always transcribed throughout the cell cycle of cycling stem cells, which has never been demonstrated. Second, it has been shown that about 10-20% of sorted X1’s could not be labeled for *piwi-1* RNA in previous publications (Reddien, 2005, and Eisenhoffer, 2008), which may skew results if those unlabeled dividing cells were still PIWI-1+.

*2) They should address how* mex3(RNAi) *is distinct from RNAi of other factors required for neoblast differentiation (i.e.* p53*). In particular, does loss of* mex3 *result in eventual loss of the neoblasts like* p53*? In*
Figure 6
*there is a ∼3 fold drop in pH3+ cells between days 12 and 15. Is this due to depletion of the neoblast pool, similar to what happens with* p53*? What about at day 18? The authors should also assess* p53 *requirement in gut/neural progenitors to determine if* mex3 *is broader in its role*.

These are good points. We believe the *mex3-1* RNAi phenotype is very different from that of *p53*. We never see loss of stem cells in the *mex3-1(RNAi)* animals as is seen after *p53* knockdown. We have added in additional lineage stains at a longer time point (day 20), as well as stains of animals with a greater number of RNAi feedings (3 feeds) to demonstrate the persistence of stem cells (Figure 4—figure supplement 1). With regards to the drop in H3P from day 12 to 15, we believe this is partly attributable to the single-feed effects of *mex3-1* RNAi actually wearing off. As seen at the day 20 lineage stains, epithelial post-mitotic progeny begin to reappear. As this rebalance of making progeny is restored (perhaps asymmetry restored to the lineage), H3P levels are seen to return close normal. Again, this is not due to a loss of stem cells as it is in *p53(RNAi)*, where the H3P levels go to zero. We attribute the fact that 100% of the animals still die to the fact that progeny have been absent for too long for the animal to recover. We have also included additional time points of H3P immunostaining (days 18 and 21), demonstrating that while *mex3-1(RNAi)* H3P levels are declining, they are still significantly increased compared to controls.

Our data also demonstrate that *mex3-1* has a broader role than *p53* in differentiation. *mex3-1* mRNA is present in all *piwi-1*^+^ and *prog-1*^+^ and *AGAT-1*^+^ cells, whereas the majority of *p53* expression is restricted to *prog-1*^+^ cells. We have included data demonstrating that *mex3-1* is also expressed in *chat*^+^ neural progenitors (Figure 6—figure supplement 1), and is not restricted to epithelial progenitors. We have also included additional data demonstrating that *p53* knockdown does not significantly reduce *ovo*^+^ and *chat*^+^ progenitors (Figure 6—figure supplement 2).

*3) The authors need to provide further quantification of the neoblast results. The model that* mex3 *is promoting the differentiation/renewal balance is influenced from their data on neoblast numbers, but at present, the data are not sufficient to strongly make the case. The EdU quantification indicated that there are more S-phase cells in* mex-3 *RNAi animals, but this is a technically limited experiment because of patchy labeling. Where the counting was done in even an individual animal would yield very different results. Furthermore, the FACs quantification of cells greater than 2C (X1s) shows no clear difference or perhaps less in* mex3 *RNAi (*Figure 4—figure supplement 1*), and no increase in expansion of neoblasts, compared to the control, in the sublethal irradiation experiment is apparent.*
Figure 5
*provides the strongest support, but no numbers are given, and it would be stronger with quantification from many animals and at multiple AP planes*.

We have provided additional quantification of neoblast numbers which we believe further support our model. We performed quantification of *piwi-1*^+^ cells in matched locations for the head, neck, and tail of animals (counting dorsal to ventral, Figure 6), as well as quantification at two more AP planes (in the middle of the pharynx, and in the tail, Figure 5—figure supplement 1), all which show that *piwi-1*^+^ cells are significantly increased after *mex3-1* RNAi. All graphs now show cell counts normalized to surface area, instead of being expressed as a ratio to controls.

With regards to the EdU quantification, our EdU labeling is fairly homogenous and similar between animals, but the protocol does result in non-homogenous background staining that is more noticeable in a single confocal plane with low magnification. We have replaced the whole-worm EdU images with new pictures containing less background. Concerning the sublethal irradiation experiment, it should be noted that *mex3-1(RNAi)* animals are more susceptible to irradiation than controls, and also die sooner than non-irradiated *mex3-1(RNAi)* animals (and not long after the day 9 post-irradiation time point). This overall decline in health may limit the ability of stem cells to expand. We do not believe that FACS-gate quantification is consistent enough to say much more than the cells are present at relatively normal levels.

*4) Along the same lines, using the PIWI antibody (*Figures 5 and 6*) labels neoblasts and their descendants. Therefore, technically, stem cell subclasses are not shown. PIWI+* hnf4*+ cells increase following* mex3 *RNAi, whereas* ovo+ *or* coe+ *PIWI+ cells decrease. How do these opposing results fit with the model? The authors should attempt to resolve quantitative impacts of mex3 RNAi on neoblast subclasses versus their non-dividing descendants using marker/*piwi*/PIWI labelings*.

We agree with the point that the PIWI-1 antibody also labels immediate neoblast descendants, and have repeated neoblast subclass quantification with *piwi-1* labeling instead (Figure 5). Our new quantification similarly shows that the three subclasses measured all significantly increase after *mex3-1* knockdown. Combined with our new data examining the ratios of *piwi-1*^+^PIWI-1^+^ cells to *piwi-1*^-^PIWI-1^+^ cells in *mex3-1(RNAi)* animals, our results show an overall increase in neoblasts and decrease in non-dividing descendants. *hnf4* was recently found to be part of an expression signature of a third subclass of neoblast (gamma subclass), and was postulated by [79] to be a gut-restricted stem cell type, not a postmitotic progenitor type. (In either case, this specific cell type has not been thoroughly investigated.) We believe the increase of *hnf4*^+^PIWI-1^+^ in *mex3-1(RNAi)* animals is due to the increase in *hnf4*^+^*piwi-1*^+^ gamma neoblasts, which we have now shown. To address the impact of *mex3-1* RNAi on gut differentiation, we have included a BrdU labeling experiment looking at entry of new cells into the gut (Figure 6) and show that it is decreased.

*5) Figures/data*:

Figure 1—figure supplement 1*: It is unclear whether the correlations were done on raw values or log-transformed values; if not log-transformed analyses, it should be plotted and analyzed this way to prevent overestimation of the correlation. Range/units on X and Y axes should be present. Graphs are overplotted, and might be better represented with 2D density plots. Red line is unclear; representing “a Pearson correlation coefficient of 1” is unclear. The correlation P values are unnecessary. RPM should be changed to CPM*.

Thank you for the suggestions, we have addressed these points in the revised figure and legend. We did not choose 2D density plots because that would show the distribution, and datasets can have the same distribution without being correlated.

Figure 1*: Y axis states “over Xins” when the comparison is to whole worms and Xins has been used to refer to a FACs gate in the literature*.

We have changed the term to “over irrad” instead of Xins and explain what that means in the legend.

*In the first paragraph of the subsection headed “RNAseq analysis of the progeny-associated X2 FACS cell fraction”: “stem cell markers can be highly expressed in the X2 fraction (*Figure 1*)” – this is not shown in*
Figure 1.

We have now plotted *piwi-1/2*, *cyclinB*, and *PCNA* on the graphs in Figure 1 and *prog-1/2* and *AGAT-1/2/3* in Figure 1.

Figure 1*: The colors are saturated, and it seems the range shown does not capture the range of the data. If they are setting a maximum/minimum for the graph that is below the range of the data this should be stated in the legend, or the shown range changed*.

We have attempted to adjust this, but it should be noted that the heatmap is meant to be a visual tool to understand the experimental paradigm of the study, not a specific way to understand the expression dynamics of 80,000+ transcripts. Therefore, we intentionally adjusted the parameters to make the plot as binary as possible to reflect the Venn diagrams. The reality is that z-score and fold-change cutoffs are arbitrary.

Figure 1*: The “Lowess” curve (Loess fit?) is not needed, and the authors are encouraged to remove it*.

We have removed these curves.

Figure 1*: “new progeny markers” is an awkward Figure label for these genes*.

We agree with this and have now changed to label to read “epithelial progenitor markers.”

Figure 1
*legend: the description is poor – it is a heatmap, Z score range, etc*.

We have added in additional information.

Figure 1—figure supplement 2
*is hard to read; trimming the end/uninformative part(s) would help focus on the region with similarity. Shown is not a “MUSCLE alignment” but a multiple sequence alignment done with MUSCLE. There is more than one shade of blue, which either represent similar or identical aa. The three rows of information below the alignment should be better described. Why not show all 24 in the alignment?*

We have made the suggested changes in the alignment (trimmed) and descriptions in the figure legend for the blue shading and data below the alignment. We are not including all 24 *prog*-homologs because they are not X2 or X1 enriched, nor are they irradiation sensitive. This is not a definitive study of this gene family. We have changed the wording of the sentence; however, it was grammatically correct as written due to the fact that the “MUS” in MUSCLE stands for “MUltiple Sequence”.

Figure 2*, genes are divided into early and late, but a given late marker in principle could also have been present in early cells. This might be worth noting in the text*.

We have performed all the pairwise dFISH and this is not the case (Figure 2—figure supplement 3). We have noted this in the text.

*In the first paragraph of the subsection headed “New progeny markers label either early or late progeny of the epithelial lineage” the authors find that 5.4% of* prog-1 *cells expressed low levels of AGAT-1, and Eisenhoffer 2008 reported 44%. They refer to these populations as “two mainly non-overlapping”, which needs some clarification*.

We have added in a sentence to clarify the discrepancy. As an interesting side note on our percentage overlap for any planarian reviewers/readers of this document, we found that there is some sort of non-specific riboprobe interaction that gives false positive results between *prog-1* and *AGAT-1*. This is only apparent using our new markers and *prog-2* along with careful analyses. Essentially, we believe that the Eisenhoffer, 2008 result is an artifact of older ISH techniques. It is also why we need as many markers as possible for any given cell type.

Figure 3*.*

*Part A might be better as time following initiation of RNAi*.

We have adjusted the graph to reflect days following RNAi initiation.

Part E: what happens at longer time points post RNAi? How much expression is not irradiation sensitive?

We have added whole-mount ISH stains of *mex3-1* at later time points after irradiation which show that very little to no expression remains by 7 days, which was already known by RNAseq at this time point.

mex3 *orthology: The tree lacks an outgroup(s) with a similar RNA-binding domain to that present in this protein family. Therefore, without outgroups, any non-mex3 family protein would give the same result as planarian MEX3 as presented in the tree in*
Figure 3—figure supplement 1*.*

An outgroup would certainly be helpful if we wanted to determine whether a given KH-domain protein was a MEX3 ortholog or closer to another family member of the KH-family. However, in this case, we have pre-selected all the genes to be of the *mex3* family, which is clear by reciprocal BLAST. Therefore, we are not trying to determine whether they are actually *mex3*-family genes, which is when outgroups are most helpful. We are only interested about the potential orthology. By selecting an outgroup, you are telling the analysis program that you know something about the ancestral sequence of the gene family in the tree, which we do not. Unrooted trees take an unbiased approach of detecting the relationships between the tips of the branches. The only question we were asking in our phylogeny is whether there is direct orthology in the multiple vertebrate sequences with the multiple planarian sequences. The ancestral condition is clearly a single *mex3* gene in invertebrates, so already by parsimony it would be unlikely that the orthology of a given planarian gene would be orthologous to an individual vertebrate gene, but it was a formal possibility to test with this analysis. For that to be true in reality, it would require gene loss of the other homologs in most other phyla, as well as the genes not to have been duplicated in the whole genome duplications vertebrates have undergone (i.e. not parsimonious). Our phylogeny simply confirms that the *mex3* planarian genes are planarian paralogs. We are happy to remove the phylogeny if necessary, but do not agree with trying to outgroup the gene family. The downsides of outgrouping are well documented: “The problem of rooting rapid radiations. ”MolBiolEvol*.* 2007 or “Outgroup misplacement and phylogenetic inaccuracy under a molecular clock – a simulation study” Syst Biol*.* 2003.

Figure 4*.*

*C: there is a clear TUNEL effect, without clear explanation. It might be worth simply noting in the text that the explanation for this late rise in TUNEL+ cell number is unclear*.

We apologize for this oversight and attribute the rise in TUNEL to decline in overall animal health and when lesions begin to appear, etc. (although lesioned animals were not stained). The text has been modified to state this.

*C/D: Where counts were performed should be stated*.

Whole-animal counts were performed. This has now been noted in the figure legend.

*The X2s by FACs don't seem to be decreased following mex RNAi (*Figure 4—figure supplement 1*), and yet there is a strong impact on* prog*-like+ cells by in situ. Is there an explanation? For example, is the fraction of X2s that are neoblasts (*piwi+*) increasing?*

We do not provide interpretation for this result. Because prog and AGAT cells only make up ∼20% of the X2 gate, and because there are stem cells in the X2 gate, we believe there is not a strong impact of losing progeny and increasing stem cells on the relative amounts of cells in each gate. Perhaps a future experiment would be to purify the remaining X2 cells and sequence them to see what transcripts are expressed in this gate. As stated above, we put relatively little weight on results of FACS gate percentages due to high variability in the technique currently based solely on vital dye staining.

Figure 5*.*

*B*: *This is a nice result clearly showing neoblast increases can occur in* mex3 *RNAi animals. The images are dark and hard to see in the present figure file.*

Thank you, we have made sure the brightness is balanced, but believe this was a file conversion issue as well in the original submission.

*D: Using the PIWI antibody labels neoblasts and their descendants. Therefore, technically stem cell subclasses are not shown. I recommend having a simple cell count plot instead of showing ratios. The high variance of the data in the* hnf4 *and* zfp1 *counts, suggest the the* p*-value is a bit surprising, although without seeing the counts, that is hard to evaluate*.

As mentioned above, we have repeated the experiment using a *piwi-1* riboprobe instead of antibody, to more accurately label neoblasts. We have also changed the graph to show counts normalized to area (the same area was counted for all worms), instead of ratios normalized to control RNAi animals. A scatter plot of raw counts on piwi-1/PIWI-1 ratios of cells is now in Figure 6 and should be clear.

Figure 6*.*

*A: There wasn't really a “chase”, it was a pulse + 7 days. The wording could be modified*.

We have modified the wording accordingly.

*C: See comments on*
Figure 5*. The PIWI ab here is less of a concern, if the wording is clear that these are neoblasts and/or descendants involved in tissue turnover. The ratio comment applies here as well. Number of animals counted should be noted.*

We have replaced the bar graphs to show counts instead of ratios, and have included number of animals counted in the figure legend.

*Is* mex-3 *expressed in the any/all of these three sets of cells?*

We have examined *mex3-1* expression in *chat*^+^PIWI-1^+^ progenitors and found that almost all (97.8%) express *mex3-1* (Figure 6—figure supplement 1).

Figure 7*.*

B: Are these indeed a similar class of cells studied in prior parts of the paper? (Several double-FISH tests would quickly confirm this). How do these look in the WT-high X-low data?

We have examined expression of two of these genes by irradiation and dFISH. Both show kinetics of down-regulation similar to *AGAT-1*, and co-localize highly with *AGAT-1* (Figure 7—figure supplement 2). All 21 genes can be categorized as having a WT^high^X^low^ expression profile, which we have added to the text.

C: numbers, rather than ratios would give reader a better understanding of how many cells were present/ counted. What fraction of these are PIWI+?

We have replaced the bar graph to show number of cells counted instead of ratios. We have also indicated in the top right corner of the *SmedASXL059179*/PIWI-1 image panel the percentage of *SmedASXL059179*^+^ cells that express PIWI-1.

What is the expression of the candidate pharynx progenitor marker in irradiated worms?

It is progressively lost and completely gone by day 7 post-irradiation. This has now been added to Figure 7—figure supplement 2.

[Editors' note: further revisions were requested prior to acceptance, as described below.]

*1) Reply to point #2: “We have included data demonstrating that* mex3-1 *is also expressed in* chat^+^
*neural progenitors (*Figure 6—figure supplement 1*)”. This was done with PIWI1 antibody and therefore these cells are not necessarily progenitors; they could be newly differentiated cells.*

We believe this is confusion in terminology. By “progenitors” we do not mean an equivalent of dividing progenitors or transit amplifying cells from other systems, but instead imply a committed cell that has not yet fully differentiated and physically integrated into the terminal tissue, similar to the DCX^+^ non-dividing “neuroblasts” of the mammalian cortex. To try to minimize this confusion, we named most of our newly discovered transcripts in the “postmitotic progeny” series. In this current revision, we have also tried to minimize the use of the word progenitor. So to directly address this comment: “progenitors”, “stem cell progeny”, and “newly differentiating cells” are equivalent.

Because PIWI-1 labels stem cells and immediate post-mitotic descendants, we consider “progenitors” to be cells that are PIWI-1^+^[tissue-specific marker]^+^, to mark cells in the early stages of commitment. For some tissues like the eyes, progenitors are easily demarcated as a trail of *sp6-9*^+^ and *ovo*^+^ posterior to the eyes. These trail cells are PIWI-1^+^, but can also be counted without the use of PIWI-1^+^ labeling. For tissues without clear physical separation of progenitors, like neural tissues, the use of PIWI-1^+^ is needed. Additionally, not all progenitors express *piwi-1* mRNA; for example, in the trail of eye progenitors, *piwi-1* mRNA expression is limited to the posterior-most cells. As no permanent lineage tracing can yet be done in our system, it is impossible to know the terminal fates of individual cells and whether these *piwi-1*^+^ progenitors self-renew. We have attempted to clarify this in the text, and this rebuttal document will also help readers.

*“*p53 *knockdown does not significantly reduce* ovo^+^
*and* chat^+^
*progenitors” – again, it was done with a PIWI1 antibody, and therefore those are not necessarily progenitors.*

*The wording should be addressed accordingly (or* smedwi *mRNA could be used).*

We have clarified in the text our use of the term “progenitor”. We use the same progenitor term used in Lapan and Reddien (Plos Genetics 2011) and Lapan and Reddien (Cell Reports 2012) describing the *sp6-9*^+^ and *ovo*^+^ cells as eye progenitors, respectively. A similar terminology was used in the *eLife* manuscript by [3] regarding *FoxA*^+^ progenitors. Text changes can be found in the third paragraph of the Introduction and in the subsection headed “*mex3-1* is required for turnover of multiple tissues”.

*Related to this, in the reply to major point 4: Would the authors’ argument about* hnf4 *predict that* smedwi *(mRNA)+* coe*+ cell would go up in frequency? This could in principle be tested to further support the proposed model*.

Considering that we do not know whether a neural stem cell exists that is akin to gamma, zeta, or sigma neoblasts, nor what terminal cell types would be made by such a neural stem cell, this would be an over-interpretation. The manuscript is not focused on stem cell types, but is focused on how non-stem cell fates are made. Our model is currently supported by measuring gamma, zeta, and sigma neoblasts, as well as counting thousands of *piwi-1*^+^PIWI-1^+^ neoblasts and *piwi-1*^-^PIWI-1^+^ immediate post-mitotic descendants as requested previously. The result in Figure 6 where PIWI+coe+ cells in between the brain lobes are lost suggests these are progeny, not a stem cell class.

*2)*
Figure 1—figure supplement 1*. Again, in this reviewer's view plots should be log transformed and correlation done on log-transformed values*.

This figure has been updated, resulting in no change in interpretation.

*3)*
Figure 1*. The response did not satisfactorily address the review comment. The legend states the z-score range shown, but if the data is not showing the actual z-score values*, *why not state what the actual range of the z-scores was in the data in the legend such that it is clear that some values were much lower/higher than the cutoffs on the range shown?*

We have modified the figure legend to more accurately state that the heatmap shows transformed z-scores, not raw z-scores. We used the Partek software suite to construct our heatmap, to allow us to adjust the saturation range of z-scores in order to make the heatmap more interpretable for the reader as a visual tool. The saturation range for z-scores is more or less arbitrarily picked to increase the resolution of expression differences and de-emphasize the saturated signals. A transformed range of z-scores from -0.5 to 0.5 empirically yielded the best results. We have provided supplemental files of all the raw transcript counts that went into the heatmap, and interested readers can peruse further if they desire more detail on the maximal/minimal fold changes and z-ranges using their preferred method. All significantly enriched transcripts by DESeq are shown in the log-fold change MA plots in Figure 1.

*4) MEX-3 orthology: The authors clarified their goal for the MEX3 phylogeny, which makes this tree of limited value; however, the answer leaves the evidence for MEX-3 orthology poorly described in the manuscript. Perhaps the authors can more explicitly describe in the methods the evidence that this is a MEX-3 ortholog (e.g., phylogenetics, unique domain architecture specific to this family, or lack of any other proteins with similarity near that present with the MEX-3 family in reciprocal genome searches)*.

We have provided more textual information about the reciprocal blasting and domains in determining MEX-3 orthology (in the second paragraph of the subsection headed “RNAi screening identifies *mex3-1* as candidate regulator of differentiation”).

[Editors' note: final revisions were requested prior to acceptance, as described below.]

Reviewer #2:

*Below is a discussion on comments regarding SMEDWI*^*+*^
*cells and the term “progenitor”*.

*[…] 5) Original Review: “*p53 *knockdown does not significantly reduce* ovo^+^
*and* chat^+^
*progenitors” – again, it was done with a PIWI1 antibody, and therefore those are not necessarily progenitors. The wording should be addressed accordingly (or* smedwi *mRNA could be used)*.

*Author Comment: “We have clarified in the text our use of the term ‘progenitor’. We use the same progenitor term used in Lapan and Reddien (Plos Genetics 2011) and Lapan and Reddien (Cell Reports 2012) describing the* sp6-9^+^
*and* ovo^+^
*cells as eye progenitors, respectively. A similar terminology was used in the* eLife *manuscript by*
[3]
*regarding* FoxA^+^
*progenitors. Text changes can be found in the third paragraph of the Introduction and in the subsection headed “*mex3-1 *is required for turnover of multiple tissues”*.

*Reviewer Reply: In the case of the eye, cells outside of the eye (an epithelial optic cup and a neural ganglion) could be visualized and described as progenitors. A SMEDWI*^*+*^
*neuron in the eye would not considered a progenitor, but an eye cell – a similar such cell would be harder to identify in a less discrete tissue/cell population.*

Comments 1-5 are all addressed by addressing Reviewer Reply Point #4.

We have removed the use of the term “progenitor” as suggested specifically for *chat*^*+*^PIWI-1^+^ and *six1/2-2*^*+*^PIWI-1^+^, and have renamed these cells “neoblast progeny”, and stated that this includes “undifferentiated progenitors and newly differentiating cells”. We do not believe these cells can be considered fully differentiated, due to their expression of the stem cell marker PIWI-1, being irradiation-sensitive, and not being stable in their gene expression as you would expect of a truly differentiated cell. (Fully mature neurons do not express PIWI-1 and are not irradiation-sensitive like this population. Guo et al*.* (2006) demonstrated that all PIWI-1^+^ cells are ablated 3 days after irradiation.). We also added a sentence about PIWI-1^+^ cells being of unknown stage of differentiation (“It is unknown how differentiated these *piwi-1-* PIWI-1^+^ cells are, but they are clearly in a transition from a stem cell gene expression state to various 12 postmitotic fates”).

Furthermore, it cannot be concluded from the [83] study cited by the reviewer that SMEDWI-1+*chat+* cells (which the paper states are “enriched in brain regions and often adopted a non-neoblast cell morphology that included long axon-like cytoplasmic processes”) are “fully integrated into tissues” as the reviewer suggests. Even the authors of that paper refer to these cells as “differentiating”, not “differentiated”. Similarly, in Scimone et al*.* (2011), SMEDWI-1^+^ cells which co-express both a transcription factor (such as six1/2-2) and a differentiated marker (such as CA or cubulin) were referred to as “cells in intermediate stages between precursors and differentiated cells”.

It is inconsistent that this reviewer will allow analogous cells for other tissues *FoxA*^*+*^PIWI-1^+^ cells, *ovo*^*+*^PIWI-1^+^ cells, and *epithelial-progeny-marker*^*+*^PIWI-1^+^ cells being called progenitors, whereas the seemingly equivalent cells in the brain or protonephridia require a different terminology.